

# Acetone-CO enhancement ratios in the upper troposphere based on 7 years of CARIBIC data: New insights and estimates of regional acetone fluxes

Garlich Fischbeck[1], Harald Bönisch[1], Marco Neumaier[1], Carl Brenninkmeijer[2], Johannes Orphal[1], Joel Brito[3], Julia Becker[1], Detlev Sprung[4], Peter van Velthoven[5], Andreas Zahn[1]

[1]Karlsruhe Institute of Technology (KIT), Institute of Meteorology and Climate Research (IMK-ASF), Karlsruhe, Germany
[2]Max Planck Institute for Chemistry (MPIC), Air Chemistry Division, Mainz, Germany
[3]Laboratory for Meteorological Physics (LaMP), University Blaise Pascal, Aubière, France
[4]Fraunhofer Institute of Optronics, System Technologies and Image Exploitation (IOSB), Ettlingen, Germany
[5]Royal Netherlands Meteorological Institute (KNMI), De Bilt, the Netherlands

*Correspondence to*: Marco Neumaier (marco.neumaier@kit.edu)

**Abstract.** Acetone and carbon monoxide (CO) are two important trace gases controlling the oxidation capacity of the troposphere. Enhancement ratios (EnRs) are useful to assess their sources and fate between emission and sampling, especially in pollution plumes. In this study, we focus on in-situ data from the upper troposphere recorded by the passenger aircraft based IAGOS-CARIBIC observatory over the periods 2006-2008 and 2012-2015. This data set is used to investigate the seasonal and spatial variation of acetone-CO-EnRs. Furthermore, we utilize a box model accounting for dilution, chemical degradation and secondary production of acetone from precursors. In former studies, increasing acetone-CO-EnRs in a plume were associated with secondary production of acetone. Results of our box model question this common presumption and show increases of acetone-CO-EnR over time without taking secondary production of acetone into account. The temporal evolution of EnRs in the upper troposphere, especially in summer, is not negligible and impedes the interpretation of EnRs as means for partitioning of acetone and CO sources in the boundary layer. In order to ensure that CARIBIC EnRs represent signatures of source regions with only small influences by dilution and chemistry, we limit our analysis to temporal and spatial coherent events of high CO enhancement. We mainly focus on North America and Southeast Asia, because of their different mix of pollutant sources and the good data coverage. For both regions, we find the expected seasonal variation in acetone-CO-EnRs with maxima in summer, but with higher amplitude over North America. We derive mean (± standard deviation) annual acetone fluxes of $(53 \pm 27) \, 10^{-13} \, \text{kg m}^{-2} \text{s}^{-1}$ and $(186 \pm 81) \, 10^{-13} \, \text{kg m}^{-2} \text{s}^{-1}$ for North America and Southeast Asia, respectively. The derived flux for North America is consistent with the inventories, whereas Southeast Asia acetone emissions appear to be underestimated by the inventories.



## 1 Introduction

Acetone ($CH_3COCH_3$) is the most abundant small ketone in the upper troposphere (UT) with mixing ratios occasionally exceeding 2 ppb in summer (Singh et al., 1994; Pöschl et al., 2001; own measurements). In the dry UT, acetone constitutes an important source of $HO_x$ radicals and ozone (e.g. Singh et al., 1995; McKeen et al., 1997; Folkins and Chatfield, 2000;

Neumaier et al., 2014). At high $NO_x$ levels, acetone can form peroxyacetyl nitrate (PAN), which acts as a temporary reservoir for $NO_x$ thus enabling long-range transport of reactive nitrogen (Singh et al., 1986, 1992; Folkins and Chatfield, 2000; Hansel and Wisthaler, 2000; Fischer et al., 2014). Consequently, acetone is considered to be a key species in the chemistry of the upper troposphere and lower stratosphere (UTLS) (e.g. Fischer et al., 2012; Neumaier et al., 2014).

Acetone is either directly emitted by anthropogenic and biogenic sources or formed in the atmosphere by oxidation of

precursor compounds (e.g. >C2-alkanes). Biogenic sources (including secondary production from biogenic precursors) are believed to account for ~50−70 % of the total acetone emissions (Jacob et al., 2002; Fischer et al., 2012; Hu et al., 2013; Khan et al., 2015). Until recently, propane was thought to be the dominant acetone precursor accounting for ~30 % of the total acetone budget (Fischer et al., 2012). However, the latest STOCHEM-CRI model calculations by Khan et al. (2015) suggest that oxidation of short-lived biogenic compounds such as α-pinene and β-pinene could account for more than 60 %

of atmospheric acetone with propane oxidation being much less important (~12 %). Acetone is also directly emitted from biomass burning (BB) (Holzinger et al., 1999; Holzinger et al., 2005) with an estimated contribution of ~4-10 % to the global source (Jacob et al., 2002; Singh et al., 2004; Fischer et al., 2012). The main tropospheric sinks of acetone are oxidation by OH and photolysis, with about equal importance in the mid-latitudes. The resulting overall tropospheric mean lifetime of acetone is in the range of 14-35 days (Jacob et al., 2002; Schade and Goldstein, 2006; Fischer et al., 2012; Hu et

al., 2013; Khan et al., 2015). Despite an increasing number of UT measurements of acetone (mainly from several research aircraft campaigns), it is obvious that there remained a paucity of representative data of global atmospheric acetone. To tackle this problem, efforts have been made to retrieve acetone from ACE-FTS (Coheur et al., 2007; Harrison et al., 2011; Tereszchuk et al., 2013; Dufour et al., 2016) and MIPAS satellite data (Moore et al., 2012), but the signature of acetone is hard to detect (Stiller et al., 2004; Waterfall et al., 2005) and the vertical resolution of the respective instruments limited to 2-

3 km (Moore et al., 2012; Dufour et al., 2016). Therefore, limited acetone data have been provided this way. Given the poor understanding of the oceans as an acetone reservoir (Marandino, 2005; Fischer et al., 2012; Dixon et al., 2013) and the strong temporal and spatial variability of other sources constraining the global acetone source clearly requires more extended data sets. Current global source estimates range from 42.5 Tg a$^{-1}$ (Arnold et al., 2005) to 127 Tg a$^{-1}$ (Elias et al., 2011).

In this study, we adopt the approach by Zahn et al. (2002) who identified CO-O$_3$-correlations from small to regional scales in

the CARIBIC dataset, and investigate the relationship of acetone and CO using 7 years of IAGOS-CARIBIC measurements covering large parts of the Northern Hemisphere.

The paper is organized as follows: In Section 2, we introduce the concept of enhancement ratios (EnRs). The IAGOS-CARIBIC project and the applied measurement techniques are described in the Sections 3.1 to 3.3. In Section 3.4, we





explain how to analyse the data. The emission inventories used for comparison with CARIBIC EnRs are described in Section 3.5. In Section 4.1, we use a box model to examine the temporal evolution of EnR. The results derived from the statistical analysis of the full data set are presented in Sections 4.2 and 4.3. We summarize the results and give a conclusion in the Section 5.

**2 The concept of enhancement ratios**

A powerful tool for quantifying acetone emissions is the analysis of enhancement ratios (EnRs) in plumes (e.g. Singh et al., 2004; Lai et al., 2011). The EnR is obtained by dividing the plume enhancement of a species X (above to the background) by the enhancement of another species Y (Lefer et al., 1994; Lee et al., 1997; Mauzerall et al., 1998):

$$EnR = \frac{[X]_{\text{plume}} - [X]_{\text{bgnd}}}{[Y]_{\text{plume}} - [Y]_{\text{bgnd}}}$$   (1)

For acetone, it became common practice to use carbon monoxide (CO) as a reference species, because both gases are emitted during incomplete combustion (Andreae and Merlet, 2001; Wisthaler, 2002; Greenberg et al., 2006; Warneke et al., 2011). In practice, the EnR is either determined by measuring the volume mixing ratios (VMRs) inside and outside the plume (e.g.

Simpson et al., 2011) or from continuous airborne measurements during plume passage **(see Fig. 1)** (Yokelson et al., 2013). In a scatter plot, the data points will ideally lie on the mixing line that connects the higher concentrations in the plume with the background.

When an EnR is measured at the source, it equals its molar emission ratio (ER) (Yokelson et al., 2013). Downwind the source, the EnR remains equal to the ER as long as production or removal of $X$ and $Y$ in the plume are negligible and as long

as the plume mixes in the same fixed background (Mauzerall et al., 1998; Yokelson et al., 2013). This is due to the fact that dividing the enhancement of $X$ by the enhancement of $Y$ normalizes for dilution, as both species dilute at the same rate (Akagi et al., 2012; Yokelson et al., 2013). We prefer to use EnR whenever it cannot be excluded that the ratio has changed since emission. As shown in Fig. 1, this is particularly the case for measurements in the UT. Plume air initially mixes with planetary boundary layer (PBL) air and subsequently enters the "cleaner" UT. Plume ratios observed in the UT significantly

differ from the PBL EnR value simply because the UT background has a different acetone-CO ratio as the PBL background. The most comprehensive overview of acetone-CO-EnRs to date has been given by de Reus et al. (2003) using data of five research aircraft campaigns. For each campaign, the authors split the data into measurements from the marine boundary layer (0-1 km), free troposphere (1-12.5 km) or lower stratosphere ($O_3 > 150$ ppb, CO $< 60$ ppb) and derived one EnR per layer. Please note, that in this way, data of different flights, i.e. data of "unrelated" measurements in terms of distance and time

span, were used to derive a single EnR estimate. The authors found different EnRs for the different layers, but, surprisingly, consistent values among the campaigns. Since then, EnRs have been frequently reported for individual plumes and various





conditions. In **Tables 1-3**, we give an overview of literature acetone-CO-ERs and EnRs, without any claim to completeness. It is worth noting that ERs are only available for biomass and biofuel burning and are generally lower (mean: 2.5 ppt ppb$^{-1}$) than the observed free-tropospheric EnRs, which are on average 9.9 ppt ppb$^{-1}$ for biomass burning plumes and 12.5 ppt ppb$^{-1}$ for other plumes. In order to understand the underlying processes that change EnR, it is worth estimating how fast plumes usually mix with background air masses. In simple models, this mixing is prescribed with a constant dilution rate. In a few studies, dilution rates were determined experimentally; the results are summarized in **Table 4**.

## 3 Methods

### 3.1 IAGOS-CARIBIC Project

In the CARIBIC project (Civil Aircraft for the Regular Investigation of the atmosphere based on an Instrument Container), regular atmospheric measurements are conducted on board a commercial passenger aircraft (Brenninkmeijer et al., 2007). The present aircraft is a Lufthansa Airbus A340-600 equipped with a multi-line air inlet system (installed below the forward cargo bay) to supply the instruments with sample air. Currently, 15 instruments for in-situ and one for remote-sensing measurements of trace gases and aerosols as well as sample collecting systems for trace gas and aerosol are installed in a modified airfreight container (1.6 ton). Since May 2005, the CARIBIC laboratory is monthly deployed during regular service for 4-6 consecutive long-range flights. Detailed meteorological analysis for the CARIBIC flights (including backward and forward trajectories) is based on ECMWF (European Centre for Medium range Weather Forecasts) model data and provided by van Velthoven (2016). In 2008, CARIBIC joined the European Research Infrastructure IAGOS (In-service Aircraft for a Global Observing System) and is named IAGOS-CARIBIC since then (Petzold et al., 2015). In April 2015, the coordination and operation of CARIBIC was handed over from the Max-Planck-Institute for Chemistry (MPIC) to the Karlsruhe Institute of Technology (KIT). Further information about the project, flight routes and data access can be obtained from the regularly updated project website (www.caribic‑atmospheric.com).

### 3.2 Acetone measurements

In IAGOS-CARIBIC, we use a proton transfer reaction mass spectrometer (PTR-MS) for the detection of acetone and other volatile organic compounds (VOCs), e.g. acetonitrile (Sprung and Zahn, 2010). Here, we briefly describe the PTR-MS and refer to the extensive literature for details (e.g. Lindinger et al., 1998; de Gouw and Warneke, 2007). In general, a PTR-MS consists of an ion source, a reaction chamber, a mass analyser and a detection unit. In the ion source, $H_3O^+$ ions are produced and injected into a drift tube (= reaction chamber), which is continuously flushed with sample air and where $H_3O^+$ ions react with VOCs in the sample via the following reaction:

$$VOC + H_3O^+ \xrightarrow{k_{VOC}} VOC-H^+ + H_2O \tag{2}$$



The reaction takes place with the compound related collision rate $k_{VOC}$ if the proton affinity (PA) of the VOC (for acetone $PA_{Ac} = 812$ kJ mol$^{-1}$) is higher than $PA_{H_2O}$ ($\approx 697$ kJ mol$^{-1}$). The protonated VOCs and the remaining primary ions are guided by an electrical field towards the end of the drift tube and further to a quadrupole mass analyser. Protonated acetone is detected at the mass-to-charge ratio m/z 59. Since isobaric compounds are not separated with this technique, an unambiguous assignment to specific compounds is not always possible. In principle, the signal at m/z 59 may also have contributions from protonated propanal and glyoxal. However, other studies have shown that the contribution of propanal and glyoxal are negligible compared to acetone in free tropospheric measurements (Warneke et al., 2003; de Gouw and Warneke, 2007).

The ratio of the VOC-H$^+$ and H$_3$O$^+$ count rates (given in cps = counts per second) is proportional to the VMR of the respective VOC in the sample air. As the H$_3$O$^+$ count rate varies over longer time periods, the proportional factor and the count rates are normalized to $10^6$ primary ion counts (ncps = normalized counts per second). The proportional factor, known as sensitivity, is regularly derived in the laboratory by sampling a calibration gas with certified VOC concentrations (Apel-Riemer Environmental, Inc., Colorado, USA) under similar experimental conditions as during flight. The precision of the acetone measurement is mainly determined by counting statistics (de Gouw et al., 2003) and can be expressed as

$$\Delta[Ac]_{ppb} = \frac{1}{S_{Ac} \cdot [H_3O^+]_{10^6 cps}} \sqrt{\frac{[Ac]_{cps}^{sample} + [Ac]_{cps}^{bgnd}}{t_{dwell}}} \tag{3}$$

with $S_{Ac}$ the sensitivity of acetone, $t_{dwell}$ the dwelltime of the measurement at m/z = 59 and [Ac]$^{bgnd}$ the count rate measured at m/z = 59 in the absence of acetone. With a mean observed sensitivity of 30 ncps ppb$^{-1}$, a mean primary ion signal of $6 \cdot 10^6$ cps, a dwelltime of 5 seconds and a mean background signal of 60 cps, the precision is ~3-5 % at typical acetone VMRs of 0.5-2 ppb. Since 2010 the noise is on average ~2 times higher than noise derived from counting statistics in Eq. (3) due to imperfect electrical grounding.

The chemical background determines the limit of detection, which corresponds to a signal-to-noise ratio of 3 and is ~140 ppt for acetone at 5 s integration time. The accuracy is limited largely by the uncertainty of the concentration in the calibration gas, which is given as ±5 % by the manufacturer. The CARIBIC PTR-MS runs in the multiple ion detection mode and scans 16 masses within a duty cycle of 30 seconds, corresponding to 7.5 km flight distance at cruising speed. Background measurements are conducted every 50 minutes by diverting the sample flow for 5 minutes through a catalytic converter filled with a Pt catalyst (Shimadzu Corp., Japan) kept at 350° C.



### 3.3 Carbon monoxide and ozone measurements

Carbon monoxide (CO) is measured with a vacuum ultraviolet (UV) resonance fluorescence instrument (Scharffe et al., 2012) with a time resolution of 1 Hz. The CO molecule absorbs photons from a UV lamp (143 -155 nm) and emits fluorescence light over the spectral range of 150-220 nm. The number of fluorescence photons, being proportional to the CO concentration, is detected with a photomultiplier. The precision of the instrument is 1-2 ppb at an integration time of 1 s (Scharffe et al., 2012). Ozone ($O_3$) is measured with a fast and precise chemiluminescence detector described in Zahn et al. (2012) and calibrated using a likewise installed UV-photometer. At typical $O_3$ mixing ratios (10-100 ppb), the precision is 0.3-1.0 % at 10 Hz.

### 3.4 Data analysis

Data from the individual IAGOS-CARIBIC instruments are combined into single "merge" files for each flight with a time binning of 10 s. Data with a sampling frequency >0.1 Hz, like the CO measurements (1 Hz), are averaged over the 10 s intervals, whereas low frequency data (<0.1 Hz), like the acetone measurements (0.03 Hz), are assigned to the corresponding 10 s interval. The correlation analysis is restricted to UT air masses. Data from ascend and descend are rarely available because of the long run-up time of the PTR-MS after take-off and an automatic equipment shutdown procedure well before landing. Stratospheric acetone-CO correlations are not well suited for our purpose to investigate source patterns, because of the long transport times. To exclude stratospheric data, we use our concomitant CARIBIC ozone data and apply the definition of the chemical tropopause as proposed by Zahn and Brenninkmeijer (2003) and Zahn et al. (2004) and verified by Thouret et al. (2006). Air masses with an ozone concentration above the threshold value of

$$[O_3]^{\text{TP}} = 97 \text{ ppb} + 26 \text{ ppb } \sin\left(2\pi \frac{doy-30}{365}\right), \tag{4}$$

where *doy* denotes day of the year, are identified as stratospheric and excluded. In the rare event of ozone data being unavailable, we use potential vorticity (PV) calculated from ECMWF model and discard measurements with a PV > 2 pvu, a threshold commonly used to define the dynamical tropopause (e.g., Hoskins et al., 1985; Holton et al., 1995). In this way, 42 % of the acetone-CO data was identified as stratospheric.

In the remaining dataset, we search for physically meaningful correlations in all possible subsets of data fulfilling the following two requirements adapted from Zahn et al. (2002) and Brito, 2012: (i) The subset consists of at least 10 successive measurements that are each other no further apart than 50 km and cover less than 500 km flight path; (ii) The range of CO VMRs in the subset is greater than 10 times the average measurement uncertainty of CO. These criteria ensure that only temporal and spatial coherent events with a "fresh" source signature are considered and will be discussed in more detail in Section 4.1. For each possible subset, Pearson's linear correlation coefficient r and corresponding p-value are calculated. We assume a good linear correlation in the event r > 0.5 and p < 0.05 (5 % significance level). In such a case, the slope is



calculated using the bivariate least-squares method of Williamson-York (York, 1966; Williamson, 1968; York et al., 2004) as suggested by Cantrell (2008). The Williamson-York fit has the advantage to account for the different uncertainties of both acetone and CO measurements and precludes a dependence of EnR on the axis assignments.

A high Pearson´s correlation coefficient can also arise when respective acetone-CO-VMRs form two clusters. To exclude

such physically meaningless correlations, we implemented a cluster analysis based on Gaussian mixture models (GMM) (Everitt and Hand, 1981; McLachlan and Peel, 2000). In our case, two GMM are fitted to the acetone-CO subset. The first model expects only one cluster and the second two clusters. In order to choose the best fitting model, we use the corrected Akaike's information criterion ($AIC_C$) (Sugiura, 1978; Hurvich and Tsai, 1989). Subsets with an $AIC_C$ suggesting two clusters ($AIC_{C,n=2} < AIC_{C,n=1}$) are discarded. **Figure 2** shows two exemplary subsets: Whereas the first subset shows no

clustering, the second is affected by a strong clustering into two groups with no measurements in between to support the correlation. Although the correlation coefficient ($r = 0.79$) suggests a good correlation, the cluster analysis reveals that two well-separated air masses were measured. Such a subset is excluded from our analysis as the above-mentioned rejection criterion is fulfilled.

In general, our approach differs from the "classical" straightforward approach in the way that the diagnosed correlations are

by definition limited to temporal and spatial coherent events. The enhancement ratios detected with our approach mainly characterize the mean partitioning of acetone and CO sources in the boundary layer on a regional scale. The spread of these source regions depends on the time the analysed air parcel spends in the boundary layer before it is released into the free and upper troposphere. Therefore, one could interpret the correlations derived from our approach as "event-based" EnRs, whereby the "event" is the release of an individual air parcel out of the boundary layer into the free troposphere. In contrast

to our analysis, non-coherent correlations detected in former studies will often mirror spatial (e.g. latitudinal) gradients of acetone and CO, respectively, or imply differences of the trace gas composition of different air masses, but not enhancement ratios that characterize pollution sources and the chemical processing between emission in the boundary layer and sampling in the upper troposphere. For this reason, we believe that our approach is best suited for the analysis of source patterns with tropospheric EnRs.

**3.5 Emission inventories**

In this study, we use surface emission data from different inventories made available in the ECCAD (Emissions of atmospheric Compounds & Compilation of Ancillary Data) database (Granier et al., 2013) of the French Atmospheric Chemistry Data Centre ESPRI (former Ether; http://eccad.sedoo.fr/). The objective is to derive the total acetone flux from the boundary layer into the upper troposphere for different regions and compare this flux with the acetone source strengths

derived from the enhancement ratios and CO inventory data. Ideally, we would have preferred to use inventory data of exactly the same years for which CARIBIC data were used in this study (2006-2008; 2012-2015), but not all data in the ECCAD database is yet available for the full period. Therefore, we chose the last 6 years with complete data coverage (2005-2010) as reference period. This period has a similar duration as the CARIBIC periods and at least covers the first CARIBIC



data period. Except for biomass burning emissions, there is currently only one possible inventory for each source type for the given period. Hence, anthropogenic emissions are taken from the MACCity inventory (van der Werf et al., 2006; Lamarque et al., 2010; Granier et al., 2011; Diehl et al., 2012) and biogenic emissions from MEGAN-MACC (Sindelarova et al., 2014). For biomass burning emissions we decided to use the GFED3 inventory (van der Werf et al., 2010) instead of GFASv1.0 for

reasons of easier data handling, as GFED3 data in ECCAD has the same temporal and spatial resolution as MACCity and MEGAN-MACC. Furthermore, Kaiser et al. (2012) found that the budgets of GFED3(.1) are consistent with GFASv1.0.

As we are interested in the total flux of acetone, i.e. primary emissions and secondary production, we also include emission data from the major precursors of acetone and CO. According to Jacob et al. (2002) and Fischer et al. (2012), the two dominant precursors of acetone are propane (13-22 Tg a$^{-1}$ acetone) and monoterpenes (5-6 Tg a$^{-1}$ acetone). In order to

estimate the acetone source from propane oxidation, we use propane emission data from MACCity and GFED3 and a molar acetone yield of 72% (Jacob et al., 2002; Pozzer et al., 2010). Approximately half of the monoterpenes in MEGAN-MACC are α- and β-pinene (Sindelarova et al., 2014), which are known precursors of acetone as well. We use molar acetone yields of 15% and 16% for α- and β-pinene respectively that were reported by Wisthaler et al. (2001). For the secondary production of CO, we only consider precursors with an annual global contribution of more than 25 Tg CO according to Duncan et al.

(2007) and an atmospheric lifetime shorter than that of acetone, i.e. isoprene, methanol, monoterpenes, ($\geq C_4$)-alkanes, ($\geq C_3$)-alkenes and ethene. The respective CO production yields are taken from the same study and do not account for loss of intermediate traces gases by deposition, which might over-predict the contribution from longer-lived precursors (Duncan et al., 2007).

## 4 Results and Discussions

### 4.1 Temporal evolution of EnR between emission and sampling

For the CARIBIC measurements in the UT, it is important to consider the possible temporal evolution of the EnR, because transport timescales and typical tropospheric lifetimes of acetone and CO are of comparable range. So far, the combined influence of dilution and chemical transformation on acetone-CO EnRs has not been addressed in previous studies. In order to better assess their impact, we first examine the temporal evolution of EnRs from a theoretical point of view. We apply a

simple one-box model, in which the box represents the volume of the plume at time $t = 0$. Whereas the plume expands with time, the considered box volume is held constant to take dilution into account. The temporal evolution of the mixing ratio of a compound $X$ inside the plume can then be approximated by (McKeen and Liu, 1993; McKeen et al., 1996):

$$\frac{\mathrm{d}[X]_{\mathrm{plume}}}{\mathrm{d}t} = -L_X [X]_{\mathrm{plume}} - D\left([X]_{\mathrm{plume}} - [X]_{\mathrm{bgnd}}\right) + P_{Z,X}[Z]_{\mathrm{plume}} \tag{5}$$



where $L_X$ is the overall chemical loss rate of $X$, $D$ is the first order dilution rate and $P_{Z,X}$ is the production rate of $X$ from the oxidation of the precursor compound $Z$. The overall chemical loss rate $L_X$ is the sum of all loss mechanisms, which are for acetone reaction with OH and photolysis ($L_{Ac} = k_{Ac} [OH] + J_{Ac}$) and for CO reaction with OH ($L_{CO} = k_{CO} [OH]$). As the lifetimes of both species are at least weeks, we simply assume constant reaction and dilution rates over the considered time

5    period. Consequently, we apply daily averaged photolysis rates obtained from the tropospheric ultraviolet and visible radiation model (TUV, version 5.0; Madronich and Flocke, 1999; Madronich et al., 2010), which uses the quantum yields for acetone by Blitz et al. (2004), and monthly mean OH concentrations from Spivakovsky et al. (2000). The OH reaction rates are taken from the latest recommendations of the IUPAC (International Union of Pure and Applied Chemistry) Task Group on Atmospheric Chemical Kinetic Data Evaluation (Atkinson et al., 2004, 2006).

10   As we are interested in the evolution of $[X]_{plume}(t)$, we need to integrate Eq. (5), which is impeded by the additional time-dependent variable $[Z]_{plume}$. We show the following steps without the production term $P_{Z,X} [Z]_{plume}$ in order to derive an analytical solution. However, in the examples given in **Fig. 3**, we have solved the full equation numerically including $P_{Z,X} [Z]_{plume}$ by progressively calculating the changes of $[X]_{plume}$ and $[Z]_{plume}$ at hourly intervals, which are short compared to the net reaction rates. Integration of Eq. (5) without the production term $P_{Z,X} [Z]_{plume}$ leads to:

$$[X]_{plume}(t) = \left([X]_{plume,t=0} - \frac{D}{L_X+D} [X]_{bgnd}\right) e^{-(L_X+D)t} + \frac{D}{L_X+D} [X]_{bgnd} . \qquad (6)$$

Since the function does not consider the quasi-equilibrium of the background, it allows for unphysical low mixing ratios in the plume ($[X]_{plume} < [X]_{bgnd}$). Thus, Eq. (6) is only valid for times

$$0 < t < t_{max} = \frac{\ln\left(\frac{[X]_{plume,t=0}}{[X]_{bgnd}}\left(1+\frac{D}{L_X}\right)-\frac{D}{L_X}\right)}{L_X+D} . \qquad (7)$$

Placing Eq. (6) into the definition of EnR in Eq. (1) leads to the time-dependent function:

25   $$EnR(t) = \frac{\left([X]_{plume,t=0} - \frac{D}{L_X+D}[X]_{bgnd}\right) e^{-(L_X+D)t} - \frac{L_X}{L_X+D}[X]_{bgnd}}{\left([Y]_{plume,t=0} - \frac{D}{L_Y+D}[Y]_{bgnd}\right) e^{-(L_Y+D)t} - \frac{L_Y}{L_Y+D}[Y]_{bgnd}} . \qquad (8)$$

In the case of no chemical processing ($L_X = L_Y = 0$) or if $L_X, L_Y \ll D$, Eq. (8) simplifies to

$$EnR = \frac{[X]_{plume,t=0} - [X]_{bgnd}}{[Y]_{plume,t=0} - [Y]_{bgnd}} = const., \qquad (9)$$





i.e. in contrast to the ratio $[X]_{plume}/[Y]_{plume}$, the EnR remains constant as long as the plume mixes into the same background. In turn, any temporal change of EnR points to chemical processing inside the plume. However, as soon as chemical decomposition takes place, the assumption $L_X = L_Y = 0$ used in Eq. (9) is no longer valid and the combined impact of both chemical transformation and dilution has to be taken into account in the model.

In contrast to most previous studies, we consider both processes in our model and exclude the background reservoir from any chemical degradation (quasi-steady-state), as changes in the total balance of all sources and sinks are negligible on these short time scales. Based on the evaluation of our model, we find that the direction of change of EnR without secondary production does not only depend on the chemical lifetimes of $X$ and $Y$, as stated in former studies, but also strongly depends on the initial concentrations of $X$ and $Y$ relative to their background (cf. Eq. (8)). If the enhancement of $X$ approaches zero

faster than the enhancement of $Y$, the EnR decreases and ultimately becomes zero. For the opposite case, the EnR increases and tends towards infinity when approaching the singularity caused by the denominator.

In Fig. 3, the temporal evolution for two initial EnR values at different conditions (season, atmospheric layers and secondary production of acetone) is illustrated. The underlying mixing ratios and rates are given in **Table 5**. The free-tropospheric background concentrations are derived from CARIBIC data (see also Fig. 6). For the PBL, we use estimates based on year-

15 round measurements in Minnesota (Hu et al., 2013), California (Schade and Goldstein, 2006) and at Mace Head, Ireland (Novelli et al., 2003). The plume enhancements are scaled according to the selected EnR values. The chemical degradation rates are calculated as described above for 44°N, 1000 hPa (PBL) and 500 hPa (FT) and January (winter) and July (summer). The dilution rates are taken from Table 4 except for the FT in winter, for which we estimate a dilution rate due to the lack of available data. Propane volume mixing ratios are estimated using data of Pozzer et al. (2010), Lewis et al. (2013) and Baker

et al. (2014).

In the planetary boundary layer, EnR (without secondary production) hardly changes until dissolution of the plume, as dilution is the dominant loss process and the approximation used in Eq. (9) is valid. Taking the dilution rates (Table 5) as best estimate, our initially applied enhancements ratios will be completely dissolved in the PBL within less than one day in summer and 3 days in winter. Consequently, it is very likely that emissions of different adjacent sources may have mixed

before the release into the free troposphere. This means that the free-tropospheric EnR as observed during IAGOS-CARIBIC flights will largely reflect a mean value representing the release of regionally well-mixed PBL air into the troposphere and not the emission ratios of single point sources of acetone and CO. In other words, the mixing in the PBL ensures that air masses released into the free troposphere have a specific signature that on average represents the general proportion of acetone and CO emissions within a certain radius. As already noted in Section 3.4, the spread of this source region depends

on the residence time of the air mass in the PBL. Furthermore, the footprint is not restricted to sources that simultaneously emit both acetone and CO, but includes sources emitting only acetone or CO and also secondary production from precursors, if the residence time in the PBL is long compared to their lifetime.

As we are interested in the pure signature to assess the sources, the question arises as to how long the unaltered EnR is conserved in the free troposphere. The examples given in Fig. 3 clearly show that the EnR changes stronger and faster in



summer due to shorter lifetimes. In any case, changes become largest in aged plumes, in which the CO enhancements in the denominator of the EnR become small. As the EnR tends towards infinity when the denominator converges towards zero, the CO enhancement is more sensitive than the acetone enhancement and, therefore, better suited e.g. to define the dissolution of the plume. In Fig. 3, we use a CO enhancement of 5 ppb as dissolution criterion for the calculated evolution of EnRs. In the

5 given examples for the free troposphere in summer, the change of EnR is as high as ~300 % at the time of dissolution, strongly depending on the initial CO enhancement and the presence of secondary acetone production. As we do not have information about the actual age of the plumes observed in CARIBIC and thus cannot correct for the temporal changes, we limit our analysis to plumes with a CO enhancement greater than 10 ppb (more specifically, 10 times the mean measurement uncertainty of CO; see Section 3.4). We are aware that this threshold (open circles in Fig. 3) represents a trade-off between

10 maximizing the number of detected correlations to achieve good statistics and minimizing the consideration of aged plumes with EnRs that have been changed by chemistry and dilution to such an extent that conclusions about the source signature are not possible.

In former studies, the observation of high acetone-CO EnRs was often associated with secondary production of acetone in the plume (Wisthaler, 2002; Holzinger et al., 2005). Propane is primarily considered as precursor in this context, as it is co-

15 emitted by biomass burning and assumed to be the dominant precursor of acetone (Jacob et al., 2002; Fischer et al., 2012). If considering this source of acetone in our model, the loss of acetone is partly compensated and may lead to an increase in EnR. For plumes in the PBL, the temporal increase in EnR is therefore an indicator for secondary production of acetone. In the free troposphere, the situation is more complex and our model predicts an increase of EnR in three of four cases even without the presence of propane. Especially in summer, when the curves of the higher EnR with and without secondary

production do not differ significantly, it seems to be hardly feasible to distinguish between the different reasons of increasing EnRs.

As mentioned earlier, another reason for possible changes in EnR between emission and measurement is the subsequent mixing with different backgrounds (e.g. Mauzerall et al., 1998; Yokelson et al., 2013). Equation (8) is only valid as long as the terms $[X]_\text{bgnd}$ and $[Y]_\text{bgnd}$ are constant. Whenever the background mixing ratios change, e.g. the plume enters the free

troposphere, the EnR becomes larger under the condition

$$\left([X]_\text{bgnd,old} - [X]_\text{bgnd,new}\right) > EnR_\text{old} \left([Y]_\text{bgnd,old} - [Y]_\text{bgnd,new}\right) \qquad (10)$$

and smaller for the reverse inequality. Figure 1 illustrates this common scenario and the resulting change of the slope of the

30 mixing line.



### 4.2 Observation of EnR within IAGOS-CARIBIC

#### 4.2.1 Temporal and spatial distribution of data

The analysis of acetone-CO EnR relies on the availability of the simultaneous measurement of acetone and CO in the troposphere. At the time of the study, tropospheric acetone data is available for 105 CARIBIC flights between February 20[th], 2006 and December 13[th], 2008 and for 109 CARIBIC flights between March 6[th], 2012 and July 16[th], 2015. The gap is due to a larger modification of the instrument and subsequent re-certification. As shown in **Fig. 4**, about 90 % of simultaneous tropospheric acetone and CO measurements were carried out in the Northern Hemisphere, mainly in the subtropics and mid-latitudes along the routes between Germany and Caracas/Bogota, Sao Paolo, Chennai, Bangkok and Guangzhou/Hong Kong. Although IAGOS-CARIBIC flights to North America took place frequently, mainly stratospheric air was sampled due to the lower tropopause heights there. In order to obtain statistically reliable results, we focus on the subtropics and mid-latitudes.

#### 4.2.2 Frequency distribution of EnR

In **Fig. 5**, frequency distributions of the acetone-CO-EnR are compared for summer (JJAS) and winter (DJFM). We extended the commonly used months JJA and DJF by one month to improve the statistics. Compared to the airborne and ground observations by others (cf. Table 2-3), the CARIBIC observations provide a surprisingly clear picture. In order to quantify the distributions, we use Gaussian profiles (see parameters in **Table 6**). In winter, the approximated Gaussian profile has its centre at 8.5 ppt ppb$^{-1}$ (FMHW = 8.2 ppt ppb$^{-1}$). Thus, the centre is slightly lower than the mean literature values derived for plumes with and without biomass burning influence (9.9 ppt ppb$^{-1}$ and 12.5 ppt ppb$^{-1}$, respectively; cf. Table 2-3), but both values lie within the 1σ-range. It is clear that the real distribution differs from a normal distribution, as 33 % / 21 % of the EnRs exceed the 1σ- / 2σ-range of the Gaussian profile. This asymmetry is probably a result of the sampling of aged plumes as discussed in Section 4.1.

In summer, observed EnRs are on average ~2.3 times larger compared to winter. The centre of the Gaussian profile (19.3 ppt ppb$^{-1}$) is higher than the mean literature values, but again the values lie within the 1σ-range. The FMHW of the Gaussian profile is even ~3.7 times greater (~30.4 ppt ppb$^{-1}$), reflecting the larger natural variability in summer. As in winter, the real distribution of CARIBIC EnRs is shifted towards larger values (mean: 27.2 ppt ppb$^{-1}$). About 30 % / ~16 % of the EnRs exceed the 1σ- / 2σ-range of the Gaussian profile. The great majority of high EnRs in summer was sampled in air masses measured above or originating from North America (see next section).

To identify the reason for the considerable seasonal variation of the acetone-CO-EnR in the upper troposphere, we plot the regression lines for the mean and median parameters as derived from our EnR distributions (Table 6) alongside the VMRs of the total measurements (**Fig. 6**). It becomes clear that the factor of ~2.3 between summer and winter EnR is mainly the consequence of the considerable seasonality of acetone. The mean CO VMRs between JJAS and DJFM differ by only 6 %, simply as the CO maximum and minimum in the UT occur in March-April and September-October, respectively (Zahn et al., 2002; Zbinden et al., 2013; Petetin et al., 2015; Osman et al., 2016).





### 4.3 Regional differences in EnR and comparison with emission inventories

In this subsection, we use sample location and 5-day ECMWF backwards trajectories calculated every 3 minutes along the flight track (van Velthoven, 2016) to assign EnR to selected source regions. If a correlation is found in a subset of data (see Section 3.4), the derived EnR is assigned to each acetone-CO-data pair of the subset and to the closest 5 day back trajectory thereof. According to our box model (see grey dashed line in Fig. 3), in the free troposphere chemical decay (no dilution) does not significantly alter the EnR within 5 days; in the given examples, changes are below 5 % in summer and below 1 % in winter. Therefore, we assign each EnR to the full path of the corresponding 5-day back trajectory, which is given with a temporal resolution of 1 h. This domain-filling technique is also known as trajectory mapping and has been applied elsewhere for similar in-situ datasets (Stohl et al., 2001; Osman et al., 2016). In **Fig. 7**, the resulting geographical distribution and frequency of EnRs per 5° x 5° is shown. We are aware that back trajectories have a limited reliability. However, random trajectory errors should be negligible in our case, with respect to the large number of trajectory-mapped EnRs. In a first step, we focus on four source regions (North America, Europe, East Asia and Southeast Asia) as depicted in Fig. 7. When averaged, the EnRs are weighted according to the trajectory's residence time over the region, which should describe the situation realistically.

The mean EnR indicated in **Fig. 8** show that North America stands out with the highest EnRs observed in IAGOS-CARIBIC. In summer, the median EnR (31.7 ppt ppb$^{-1}$) is ~3.4 times larger than in winter (9.4 ppt ppb$^{-1}$) and the interquartile range is even ~5.4 times larger compared to winter. The significantly higher EnR in summer compared to winter can be explained by the following reasons: (i) the much stronger biogenic source strength in summer, (ii) the more frequent sampling of younger (acetone-rich) plumes due to strong convection and (iii) the faster increase in EnR due to shorter chemical lifetimes (see Section 4.1 and Fig. 3) The seasonality is less pronounced (in descending order) above Europe, Southeast Asia and East Asia. In contrast to the mean EnRs, individual low EnRs are observed throughout the year in all regions, as can be seen from the overlap of the lower whiskers in Fig. 8. Low EnR in summer might be an indication for rapidly ascended plumes from sources with low acetone-CO emission ratios, such as smouldering fires and other incomplete combustion processes (cf. Table 1).

### 4.3.1 Emission rates in North America

As a next logical step towards identifying the cause for the high EnR ratios over North America in summer, we consider emission estimates given in inventories for different source types (e.g. anthropogenic, biogenic and biomass burning emissions; see Section 3.5). This classification enables an assessment of the influence of the different sources on the respective total source, which helps us to interpret the observed seasonal variability in EnR. Therefore, we derive a total emission ratio (TER) defined as

$$TER = \frac{M_{CO}}{M_{Ac}} \cdot \frac{\Sigma_i S_{Ac,i}}{\Sigma_i S_{CO,i}} \qquad (11)$$




where $M$ is the molar mass of the respective compound and $S$ is the emission flux of the individual source averaged over the reference time period 2005-2010.

In **Fig. 9,** the seasonal variation of a) the acetone emission rates, b) the CO emission rates and c) the monthly means of TERs and EnRs are shown. The emissions of acetone and CO are in phase with maxima in summer and minima in winter, but the seasonal amplitude for acetone is much stronger due to the larger proportion of biogenic emissions. In Fig. 9c, we compare the inventory-based TER (with and without the consideration of biomass burning emissions) with the monthly means of IAGOS-CARIBIC EnR identified during the two time periods 2006-2008 and 2012-2015. A direct comparison only makes sense if the considered CARIBIC EnRs are not significantly altered by dilution and chemical processing. As discussed in Section 4.1, the effects of these processes are not negligible and for this reason, we limit our analysis to events with a CO enhancement of at least 10 ppb. In the ideal case, this restriction ensures that CARIBIC EnRs primarily reflect the chemical signature of the source regions.

Highest EnR are found in June and September (~40 ppt ppb$^{-1}$) with a temporary decline in-between. On the first view, this seems to be an insignificant feature, but there are some further observations that identify biomass burning as the most likely reason:

1. We observed elevated acetonitrile VMRs during this time period. In ~53 % of the air masses with correlated acetone and CO measurements we find acetonitrile VMRs greater than 200 ppt, which according to Sakamoto et al. (2015) presents a threshold for the detection of biomass burning plumes. EnRs in June appear to be unaffected by biomass burning, supported by the consistently lower acetonitrile VMR level (<200 ppt) compared to the following month.

2. The EnR decline is also apparent in TER with a shift of one month ahead, which can be attributed to biomass burning (orange diamonds in Fig. 9c). The reason lies in the low acetone-CO emission ratio of boreal forest fires of 1.6-3.0 ppt ppb$^{-1}$ (cf. Table 1). Warneke et al. (2006) found various plumes attributed to biomass burning during flights along the U.S. East Coast in July and August 2004 and concluded that 30 % of the CO enhancement is related to forest fires in Alaska and Canada, which is in good agreement with the emission inventory data (~32 %). We therefore assume that the lower EnRs in July and August (~30 ppt ppb$^{-1}$) are related to a then larger influence of biomass burning.

In July, we find a mean (± standard deviation) EnR of (28.0 ± 14.0) ppt ppb$^{-1}$, which is comparable to the ones found during aircraft campaigns over Eastern Canada, i.e. by de Reus et al. (2003) during STREAM98 in July 1998 (24.4 ppt ppb$^{-1}$) and by Singh et al. (1994) during ABLE3B in July and August 1990 (30 ppt ppb$^{-1}$). The higher variability in the IAGOS-CARIBIC EnR is presumably due to the large regional and annual variations in emissions, which are only resolved when considering local correlations over a longer time interval such as in IAGOS-CARIBIC.

### 4.3.2 Estimation of North American acetone source

Emission and enhancement ratios are frequently used to estimate global acetone emissions from biomass burning (e.g. Holzinger et al., 1999, 2005; Jacob et al., 2002; Wisthaler, 2002; Singh et al., 2004; van der Werf et al., 2010; Akagi et al.,





2011). Singh et al. (2010) denote that this top-down-approach is often useful to assess the accuracy of emission inventories that are generally derived from bottom-up data. Since we did not restrict our analysis to BB plumes, the IAGOS-CARIBIC EnRs should reflect the total acetone source. In order to derive the total acetone flux $S_{Ac}$ from our observations, the mass-corrected CARIBIC EnR is multiplied by the total flux of CO derived from inventories:

$$S_{Ac} = EnR \cdot \frac{M_{Ac}}{M_{CO}} \cdot \sum_i S_{CO,i} \ . \qquad (12)$$

For North America, we estimate a mean annual flux of $(53 \pm 27) \, 10^{-13} \, \mathrm{kg \, m^{-2} \, s^{-1}}$ corresponding to total emissions of $(6.0 \pm 3.1) \, \mathrm{Tg \, a^{-1}}$. This is in good agreement with the bottom-up estimate of $5.4 \, \mathrm{Tg \, a^{-1}}$, we derived by summing up the mean
acetone emissions given in the source-specific emission inventories (see Section 3.5).

In contrast, Hu et al. (2013) determined a North American acetone source of $10.9 \, \mathrm{Tg \, a^{-1}}$ from tall-tower measurements and inverse modelling, consisting of 5.5 Tg from biogenic sources and 5.4 Tg from anthropogenic sources. Whereas the biogenic source is similar to our estimate because the a priori source is equal (4.8 Tg), they assume a much higher anthropogenic source based on the US EPA NEI 2005 (NEI-05) inventory (12 % primary, 88 % secondary). We note that anthropogenic
emissions of acetone, propane and CO in NEI-05 are ~3, ~2 and ~1.5 times higher, respectively, than the ones given by the MACCity inventory used in this study. Several studies state that NEI-05 overestimates anthropogenic emissions of CO and other species (Brioude et al., 2011, 2013; S. Y. Kim et al., 2013; J. Li et al., 2015), whereas Stein et al. (2014) report that the anthropogenic emissions of CO in MACCity underestimate the source in Northern Hemisphere industrialized countries in winter. The latter would be in accordance with our observation of lower EnR compared to TER in winter in Fig. 9c. A larger
anthropogenic acetone source would push EnRs in the opposite direction and is not supported by IAGOS-CARIBIC EnR results. Further investigations are required to resolve the discrepancy between the above-mentioned model result of Hu et al. (2013) and the bottom-up and top-down estimates.

### 4.3.3 Emission rates in Southeast Asia

In this Section, we focus our EnR-based approach to assess regional acetone sources to the Southeast Asia region. Because
of its increasing role in global air pollution and the current shortage of in-situ studies regarding the emissions of this region, Southeast Asia stands out as a highly interesting region (Jaffe et al., 1999; de Laat et al., 2001; Lelieveld et al., 2001, 2015). The rapid industrialization is accompanied by wide-spread biomass burning resulting in a significantly different pollution source profile compared to North America (e.g. de Laat et al., 2001). Here we focus on the region of Southeast Asia (including Pakistan, India, Bangladesh, Bhutan, Myanmar, Thailand, Laos, Cambodia, Vietnam and the Philippines) as
defined in van der Werf et al. (2006). In this region, the acetone emission fluxes given in the inventories (**Fig. 10a**) are on average ~3 times higher than in North America and show a different seasonality due to the different (i.e. mainly wet tropical





and humid subtropical) climate. Emissions of CO (Fig. 10b) are mainly assigned to anthropogenic sources throughout the year, showing a maximum in March due to biomass burning emissions and a minimum in July.

In Fig. 10c, TER and IAGOS-CARIBIC EnR are plotted for comparison. As for North America, both are in the same range and show the same seasonal variation when fitting a sinusoidal function to the monthly TER and EnR, but EnR (annual mean: 12.3 ppt ppb$^{-1}$) are on average ~3 ppt ppb$^{-1}$ higher than TER (annual mean: 9.2 ppt ppb$^{-1}$). EnR values derived from the research aircraft campaign INDOEX conducted over the Indian Ocean in February-March 1999 are even higher than mean CARIBIC EnRs for February and March (9.7 ppt ppb$^{-1}$). De Reus et al. (2003) found a mean EnR of 21.6 ppt ppb$^{-1}$ and 16.2 ppt ppb$^{-1}$ when integrating over all flights in the free troposphere and in the marine boundary layer air, respectively. De Gouw et al. (2001) derived an EnR of 14 ppt ppb$^{-1}$ using data from the same campaign, but averaged acetone and CO values for level flight tracks before applying the correlation analysis. The results are consistent with the EnRs of 13.4 – 17.2 ppt ppb$^{-1}$ found in individual plumes in the marine boundary layer over the Indian Ocean (Reiner et al., 2001; Wisthaler, 2002). The reasons for the high EnRs in INDOEX compared to the mean TER of Southeast Asia (~7.7 ppt ppb$^{-1}$) and the mean CARIBIC EnR (9.7 ppt ppb$^{-1}$) can be manifold. Besides this comprehensive campaign in 1999, little data has been published on acetone emissions in this region. Based on the IAGOS-CARIBIC EnR and inventory data for CO and its precursors, we derive a mean (± standard deviation) acetone flux of $(186 \pm 81) \, 10^{-13}$ kg m$^{-2}$ s$^{-1}$ corresponding to total emissions of $(4.9 \pm 2.1)$ Tg a$^{-1}$. Langford et al. (2010) observed a mean acetone flux of $(33 \pm 181) \, 10^{-13}$ kg m$^{-2}$ s$^{-1}$ above a tropical rainforest in Malaysia in 2008, whereas Karl et al. (2004) reported a mean midday flux of $250 \, 10^{-13}$ kg m$^{-2}$ s$^{-1}$ above a tropical rainforest in Costa Rica. All three fluxes are in the same range, but hardly comparable, because of the different spatial and temporal scopes of the measurements. Whereas the in-situ flux measurements at individual locations reflect local conditions, the mean CARIBIC EnRs are representative for extended heterogeneous source regions and also capture secondary acetone production during transport. The inventory data for acetone and its precursors suggests a mean annual flux of $135 \, 10^{-13}$ kg m$^{-2}$ s$^{-1}$ and an annual source of 3.4 Tg a$^{-1}$ for Southeast Asia, which is lower than our estimates, but well within the standard deviation.

## 5 Summary and conclusions

In this study, we give a major update on enhancement ratios of acetone and CO in the upper troposphere. We present a new method to detect coherent correlations that are physically more meaningful than correlations based on spatially or temporally distant measurements. We apply this method to the IAGOS-CARIBIC dataset of acetone and CO and utilize the concept of enhancement ratios for interpretation. In former studies, free tropospheric acetone-CO enhancement ratios were often compared directly with emission ratios of individual sources, although enhancement ratios are only equivalent to the emission ratio when measured at the source. For EnRs higher than the ERs, the authors assumed secondary production of acetone in the plume. We show using a box model, that an increase in EnR is not inevitably caused by secondary production of acetone, but strongly depends on the initial quantities of acetone and CO in the plume. Dilution rates from other studies



indicate that common enhancements are rapidly mixed in the planetary boundary layer and rather contribute to the PBL background than being directly transported into the free troposphere. We conclude that an uplift of these air masses leads to tropospheric EnRs that can be seen as a chemical signature of the boundary layer air, therefore rather reflecting larger regional source patterns than distinct emissions from single point sources. As the sources vary by season, we investigate the

seasonality of EnR and find that in the Northern Hemisphere mid-latitudes they are on average 2.3 times larger in summer than in winter. Given the coverage and representativeness of the IAGOS-CARIBIC data set, it is also possible to investigate regional differences in EnR and its seasonality. We compare the seasonality of EnR observed over North America, Europe, East Asia and Southeast Asia and find the same behaviour for all four regions, but with varying degrees. We assume that these differences are mainly caused by regional differences in acetone and CO sources and therefore enable the comparison

of EnR with emission estimates of inventories. The monthly ratios of the total acetone and CO bottom-up source estimates lie well within the standard deviation of mean EnR observed over the respective region and show the same seasonal course as EnR. We calculate regional acetone fluxes by using well-constrained CO emission data and monthly averaged EnR. For North America, we estimate a mean annual acetone flux of 53 $10^{-13}$ kg m$^{-2}$ s$^{-1}$ and for Southeast Asia 186 $10^{-13}$ kg m$^{-2}$ s$^{-1}$, reflecting the dominance of biogenic acetone emissions that are larger in tropical to subtropical Southeast Asia. With our

EnR-based approach, it will be also possible to estimate regional acetone fluxes for other regions in the future. First preliminary evaluations for tropical South America show that EnRs are significantly lower than the monthly total emission ratios derived from inventories, except for months with high biomass burning emissions. It could well be that the large biogenic source of the Amazon rainforest does not provide sufficiently strong regional gradients (plumes) to be captured by our event-based detection algorithm. However, the detected EnRs might be related to biomass burning or polluted air masses

from the highly populated coastal regions. Further investigations, e.g. analysis of other tracers or evaluation of our box model adapted to the particular conditions, are necessary to understand this potential discrepancy. In addition, further measurements over this region would be of great value. We conclude that free-tropospheric EnR data with a large spatial and temporal coverage are a powerful tool to investigate the regional and seasonal differences in sources, to estimate the total acetone flux of specific regions and potentially to assess the quality of acetone emission inventories.

**Acknowledgements**

The authors wish to thank Lufthansa, Lufthansa Technik and all CARIBIC partners for their ongoing support of the IAGOS-CARIBIC laboratory. We especially acknowledge D. Scharffe for supplying the CO data and A. Rauthe-Schöch for supplying the merge files. We thank T. Gehrlein, S. Heger, C. Koeppel, A. Petrelli, D. Scharffe and S. Weber for their commitment in operating the CARIBIC container. The CARIBIC data is available from the CARIBIC website

(www.caribic-atmospheric.com) on signing the CARIBIC data protocol. We acknowledge the NCAR Atmospheric Chemistry Division (ACD) for providing the TUV Radiation Model. We thank the Centre National d'Etudes Spatiales (CNES) and the Centre National de la Recherche Scientifique (CNRS) - Institut National des Sciences de l'Univers (INSU)





for distributing the emission inventory data in the ECCAD-database. ECCAD is part of the ESPRI Data centre (fomer Ether) and the emission database of the GEIA (Global Emissions InitiAtive) project, which are gratefully acknowledged. The emission inventory data is available from the website (http://eccad.sedoo.fr/) on signing the data protocol.

**Appendix**

**A.1 Detailed description of emission data in Figure 9**

The strongest source of acetone in North America is direct biogenic emission with an annual mean of ~31 $10^{-13}$ kg m$^{-2}$ s$^{-1}$ and a strong maximum in July/August (~70 $10^{-13}$ kg m$^{-2}$ s$^{-1}$). The second largest source is secondary production from precursors. Jacob et al., 2002 and Fischer et al., 2012 assumed that propane is by far the dominant precursor of acetone on a global scale followed by monoterpenes. Here we consider propane, the monoterpenes α-pinene and β-pinene and methylbutenol (MBO)

as precursors of acetone. Methylbutenol is emitted by pine trees native exclusively to North America (e.g. Harley et al., 1998, S. Kim et al., 2010) and therefore considered as regional source. As MBO emission data are not available in the ECCAD-database, we scale the monthly monoterpene surface emissions to the total MBO source estimated by Guenther et al., 2012 and use the molar acetone yield of 0.6 given by Ferronato et al., 1998. When assuming an instantaneous conversion on ground (which is justified for the shorter-lived precursors) MBO oxidation leads to an annual mean acetone production of

~7.5 $10^{-13}$ kg m$^{-2}$ s$^{-1}$, whereas the oxidation of propane (~2.7 $10^{-13}$ kg m$^{-2}$ s$^{-1}$) and of α-pinene and β-pinene (together ~2.3 $10^{-13}$ kg m$^{-2}$ s$^{-1}$) produce considerably less acetone. Secondary production is largest in summer (~19 $10^{-13}$ kg m$^{-2}$ s$^{-1}$, ~4.1 $10^{-13}$ kg m$^{-2}$ s$^{-1}$ and ~5.7 $10^{-13}$ kg m$^{-2}$ s$^{-1}$ respectively), because of the much higher, light- and temperature-driven release of biogenic VOCs and additional propane emissions from boreal forest fires. In contrast, direct anthropogenic emissions of acetone (originating from solvent use, chemical manufacturers and car exhaust) are spread uniformly throughout the year

with ~1.6 $10^{-13}$ kg m$^{-2}$ s$^{-1}$ and account only for a small percentage of the source (~2 % in summer and ~9 % in winter). As we do not account for potential losses (e.g. deposition) of precursors before their conversion, but assume instantaneous conversion, the given emission strengths likely represent upper limits.

Carbon monoxide emissions (Fig. 9b) show a very different source composition. Direct anthropogenic emission is overall the strongest source and is evenly distributed throughout the year with a mean flux of ~4.5 $10^{-11}$ kg m$^{-2}$ s$^{-1}$. Biomass burning is

25 the second largest individual source and limited to the burning season (the summer months) with the relative source contribution exceeding 30 % and fluxes of ~9 $10^{-11}$ kg m$^{-2}$ s$^{-1}$. However, we must stress that BB emissions vary considerably from year to year and there have been years (e.g. 2004, cf. Turquety et al., 2007) in which emissions were 2-3 times higher than the average over the period 2005–2010. As already ascertained for acetone, biogenic emissions (primary and secondary) are highest in summer and account for ~45 % of the North American CO source in this season. Hudman et al., 2008 found an

30 even higher contribution of ~56 % from the oxidation of biogenic VOCs in the summer of 2004, but firstly, they limited their study to the contiguous United States, where no biomass burning occurred and secondly, considered a higher CO yield for isoprene (0.45 per C-atom, compared to 0.2 per C-atom found by Duncan et al. (2007) as global average and applied in our



study). However, we must point out that the production rate strongly depends on local $NO_x$ concentrations (Miyoshi et al., 1994) and a higher yield may be reasonable for the US. In contrast to acetone, primary biogenic CO fluxes are relatively low ($\sim$0.6 $10^{-11}$ kg m$^{-2}$ s$^{-1}$) and the secondary production from isoprene and methanol oxidation (mean annual flux: $\sim$2.2 $10^{-11}$ kg m$^{-2}$ s$^{-1}$) dominates.

## 5   A.2 Detailed description of emission data in Figure 10

Biogenic emission dominates throughout the year with a minimum in January ($\sim$61 $10^{-13}$ kg m$^{-2}$ s$^{-1}$) and a maximum in May ($\sim$148 $10^{-13}$ kg m$^{-2}$ s$^{-1}$). Biomass burning emissions peak in the late dry season (January–April) and contribute to an annual maximum of acetone emissions in March ($\sim$193 $10^{-13}$ kg m$^{-2}$ s$^{-1}$) being twice as large as the maximum acetone flux of North America.

For the same reason, CO emissions (Fig. 10b) peak in March. However, besides the large contribution of biomass burning (37 %), other anthropogenic sources make up the greatest part (48%) of the total CO flux ($\sim$131 $10^{-11}$ kg m$^{-2}$ s$^{-1}$), which is 7 times larger than the maximum CO flux of North America. Over the year as a whole, anthropogenic emissions account for $\sim$70 % of the total South East Asian CO source and are responsible for the seasonal variation with minima in summer and maxima in winter. The largest anthropogenic CO source is residential (bio-)fuel combustion for cooking and heating,

followed by emissions from the industry and transport sector (e.g. Ohara et al., 2007; M. Li et al., 2015). Biogenic emissions of CO and its precursors peak in April-May ($\sim$22 $10^{-11}$ kg m$^{-2}$ s$^{-1}$), but only account for $\sim$28 % of the total emission flux during this time and $\sim$18 % of the total annual source.

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





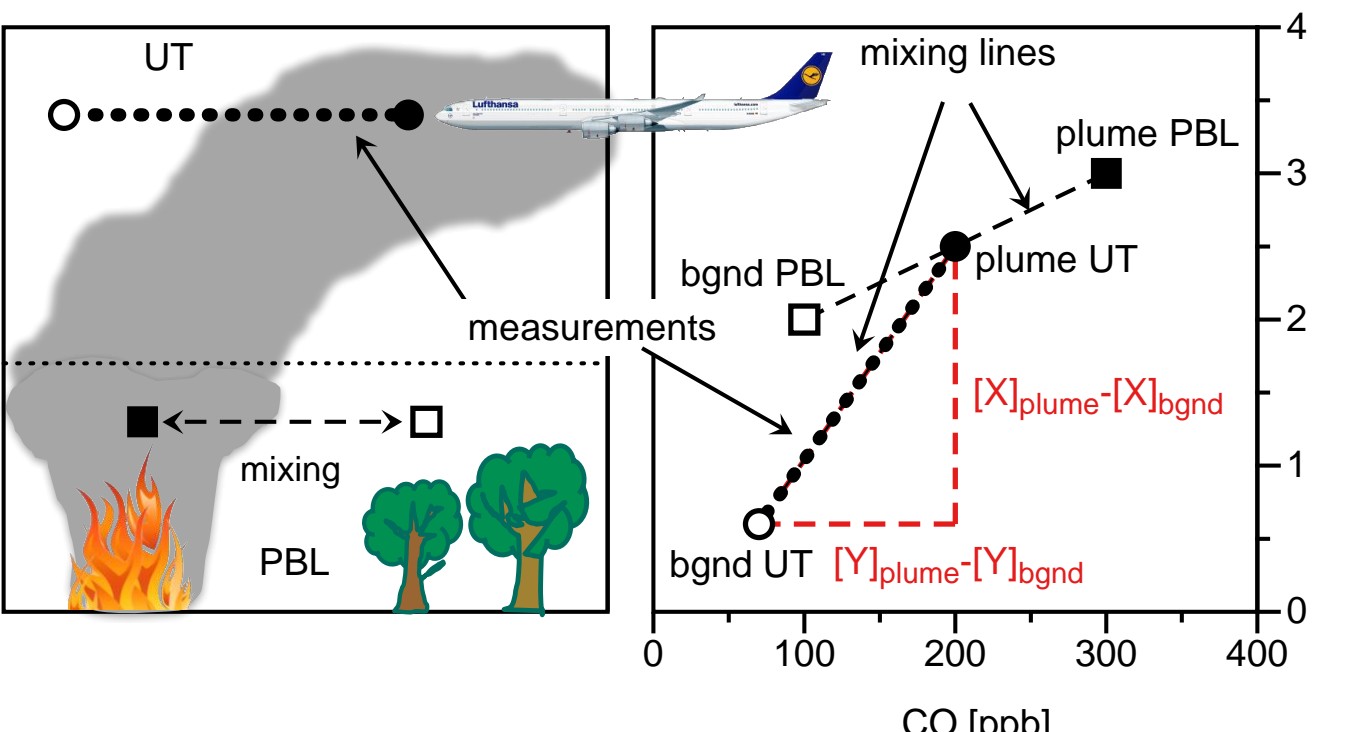

**Figure 1: Left: schematic drawing (adapted from Mauzerall et al., 1998) of a biomass burning plume and its transport from the planetary boundary layer (PBL) into the upper troposphere (UT), where sampling takes place. Right: acetone plotted versus CO concentrations. Black filled square (plume PBL): acetone/CO concentration in the fresh plume near the ground, which initially mixes with adjacent background (bgnd) air of the PBL (open square, bgnd PBL). When the resulting plume (full circle) rises into the upper troposphere, the enhancement of acetone and CO are reduced as the plume mixes with background air of the UT (open circle, bgnd UT). In an ideal case, measured concentrations lie on the UT mixing line when the aircraft passes the plume. The slope of the mixing lines, equivalent to EnR (cf. Eq. (1)), may differ considerably in the PBL and the UT.**





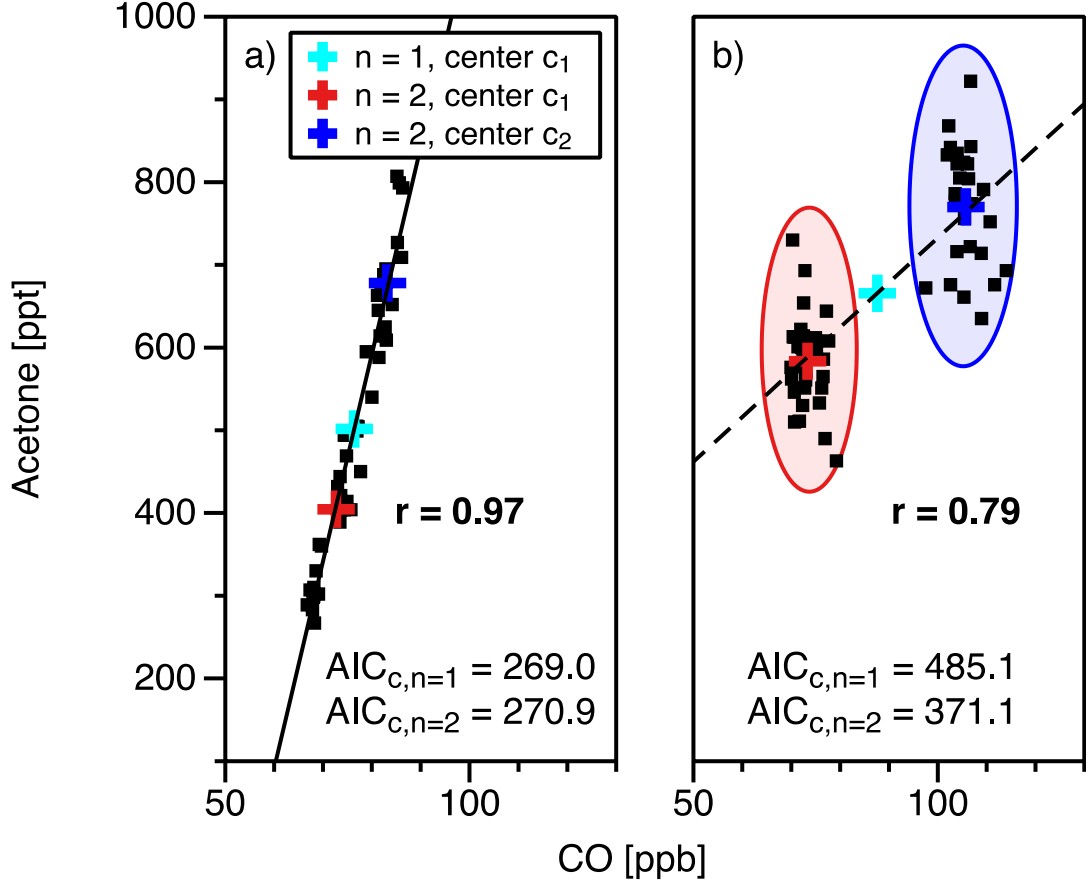

**Figure 2:** Scatter plots of two exemplary subsets of subsequently measured acetone and CO VMRs. Panel a) shows a subset with highly correlated data (r = 0.97) and no visible clustering, which is also confirmed by the cluster analysis ($AIC_{C,n=1} \leq AIC_{C,n=2}$). In Panel b) two distinct clusters are visible and automatically identified by the cluster analysis ($AIC_{C,n=1} > AIC_{C,n=2}$). Although the Pearson´s correlation coefficient indicates a good correlation (r = 0.79), this subset is rejected for determination of EnR.





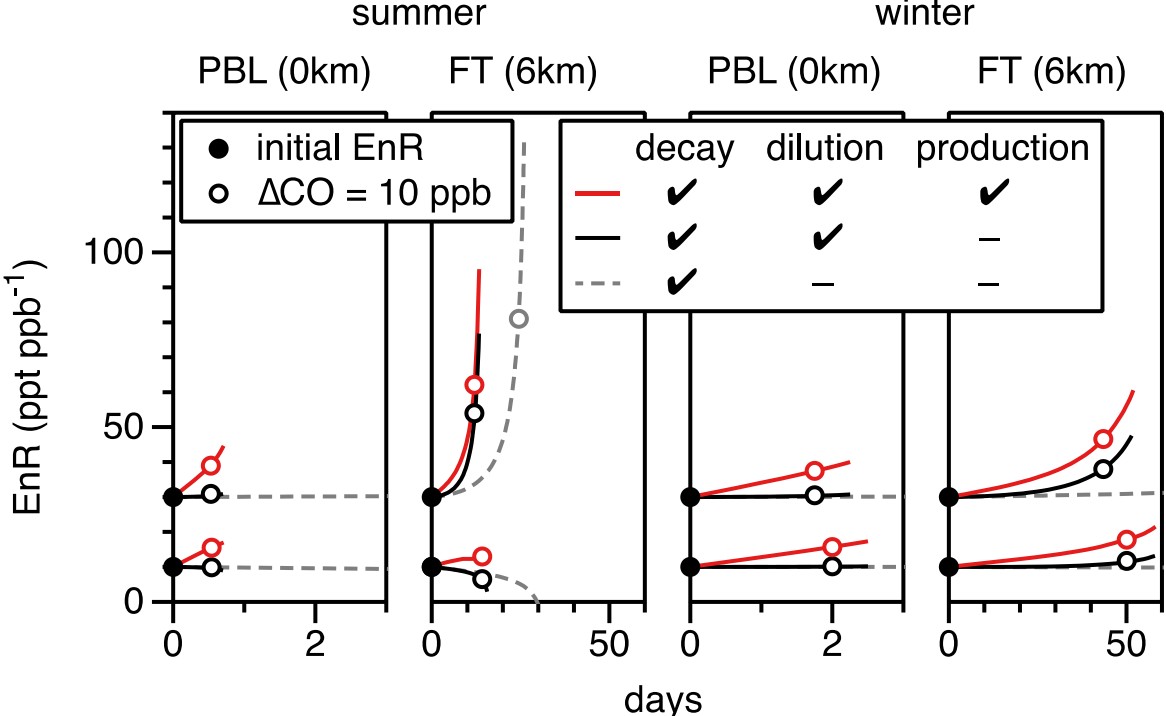

**Figure 3: Evolution of acetone-CO enhancement ratios (EnRs) in the mid-latitudinal planetary boundary layer (PBL) and free troposphere (FT) for summer and winter in the Northern Hemisphere calculated numerically (full equation including production term). In the grey dashed lines, only chemical decay of acetone and CO is considered. In the solid black lines dilution is considered in addition. The red lines represent the EnR evolution if acetone is additionally produced by the oxidation of propane. The terminating condition for the EnR calculation is a CO enhancement of 5 ppb. The open circles indicate an CO enhancement of 10 ppb. The underlying rates and concentrations are given in Table 5.**





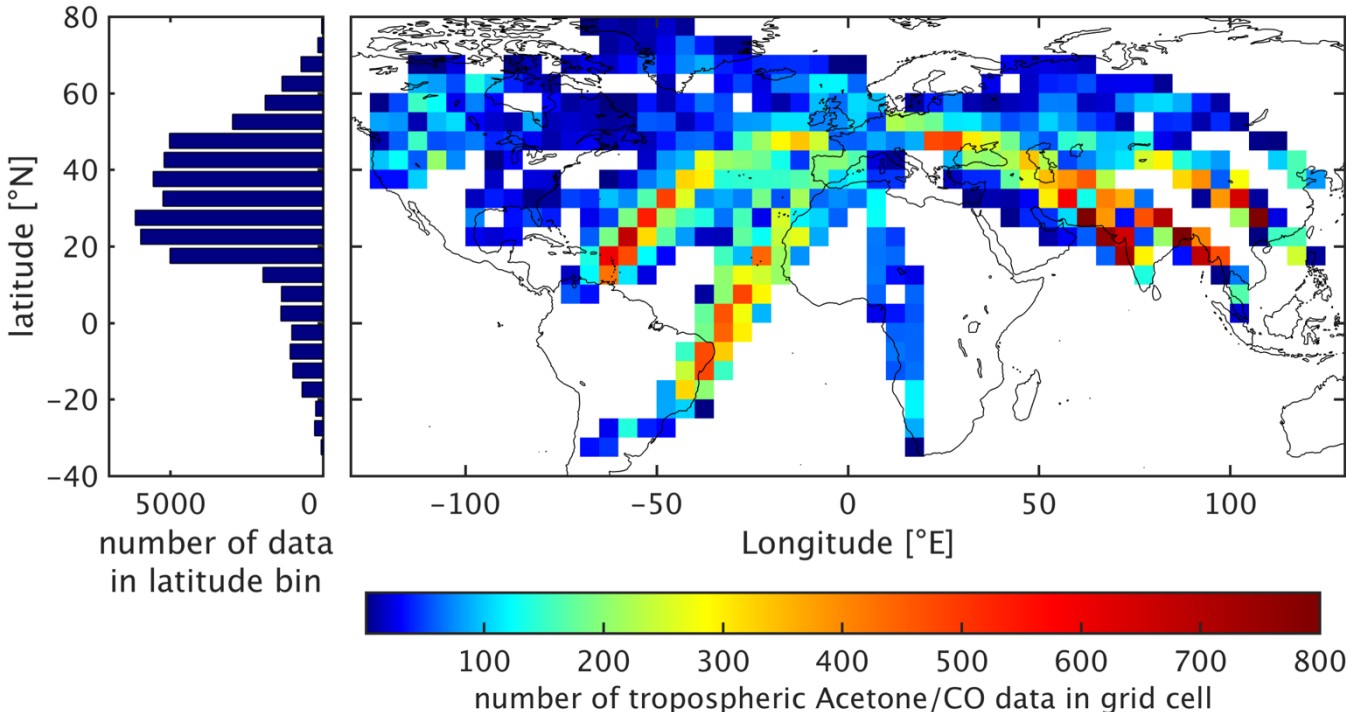

**Figure 4: Latitudinal (left) and geographical distribution (right) of simultaneous tropospheric acetone and CO measurements in the time periods 02/2006-12/2008 and 03/2012-07/2015. Grid cells without data are left bank.**



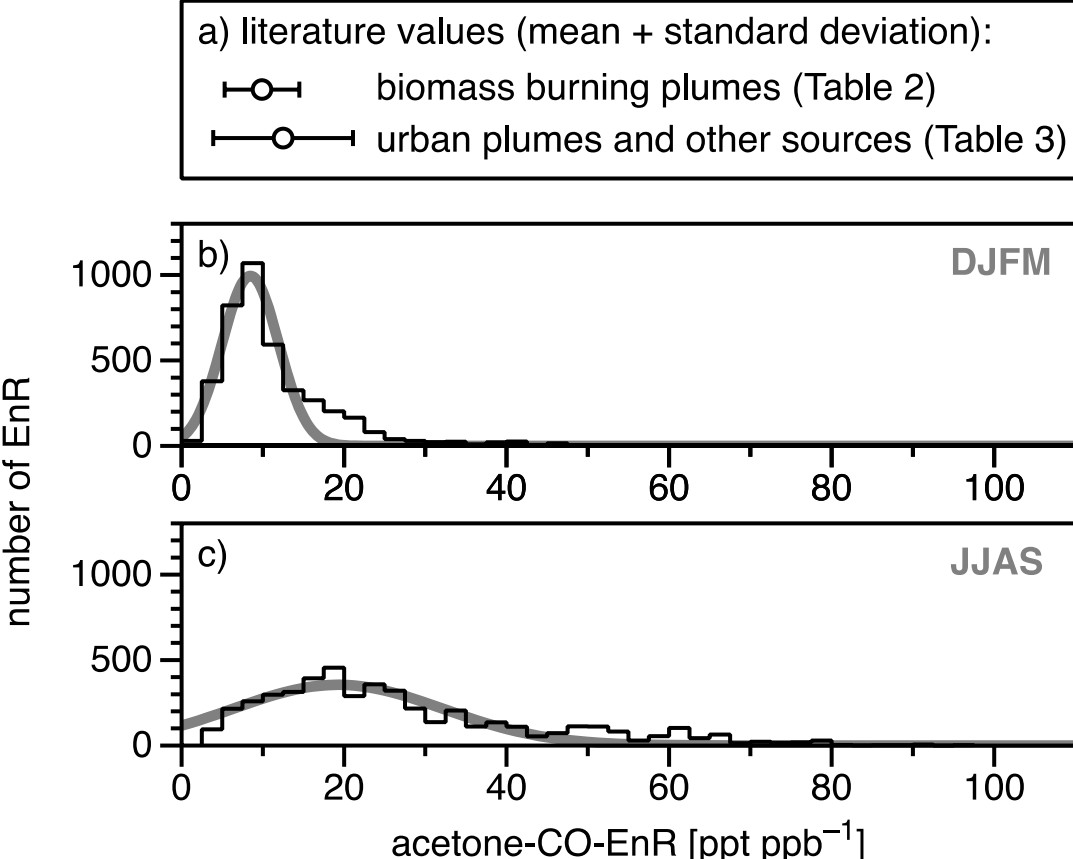

**Figure 5: Comparison of a) the mean literature values for biomass burning plumes and for plumes of other origin with the distribution of EnR observed with CARIBIC in the Northern Hemisphere subtropics and mid-latitudes (23.5°N – 66.5°N) in b) winter (DJFM) and c) summer (JJAS). The grey lines represent Gaussian curves fitted to the histograms. The values of the most important statistical variables describing the distributions are given in Table 6.**



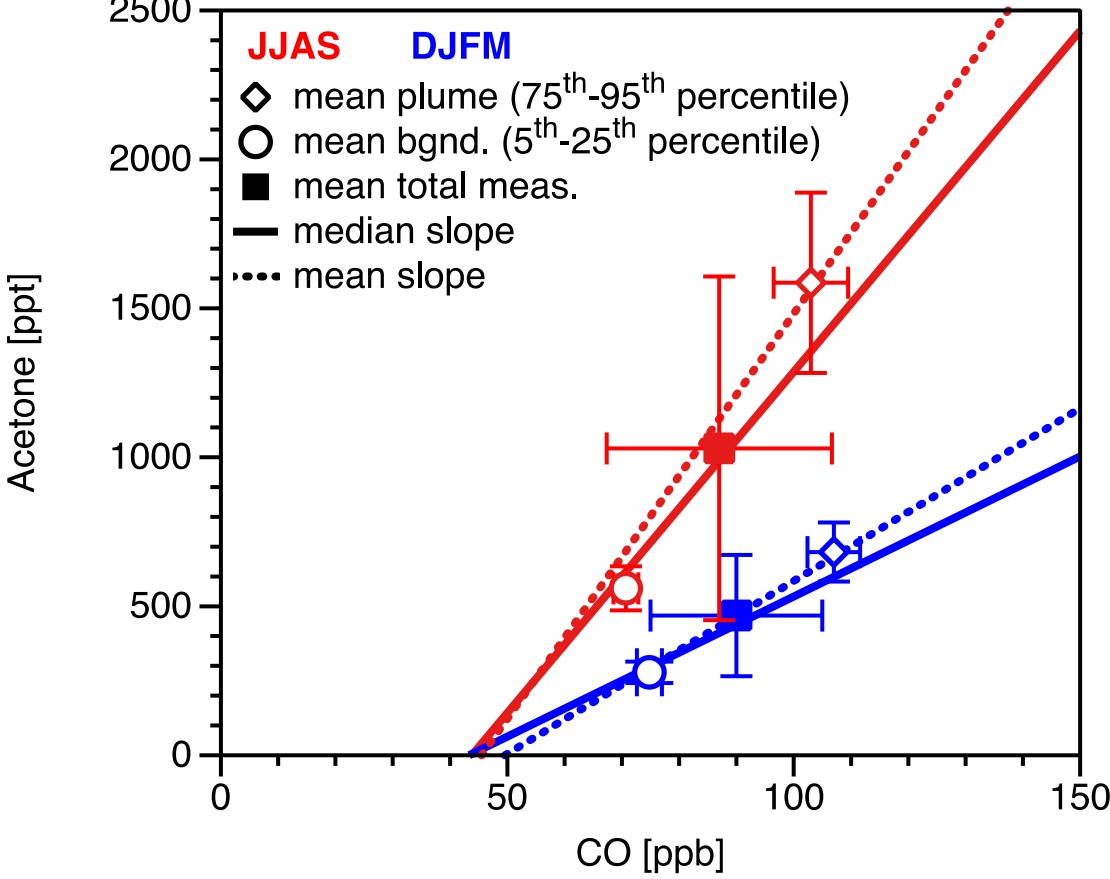

**Figure 6: Regression lines for summer (red) and winter (blue), using mean slope (solid line) and median slope (dotted line) given in Table 6. The filled squares represent the mean VMRs of the total measurements, whereas the open circles (diamonds) are the mean of all values lying in the 5th to 25th (75th and 95th) percentile, respectively.**





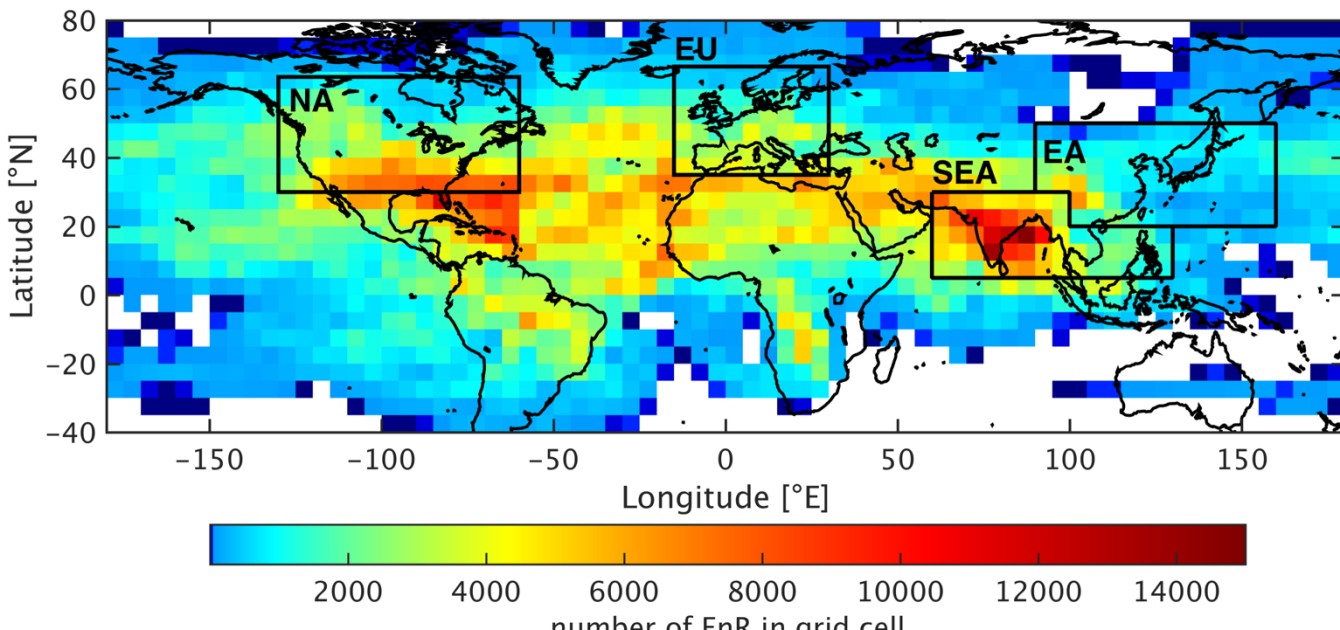

**Figure 7: Geographical distribution of EnRs attached to 5 day back trajectories. Grid cells without data are left bank. Four areas of interest (rectangles) are considered: North America (NA), Europe (EU), Southeast Asia (SEA) or East Asia (EA).**





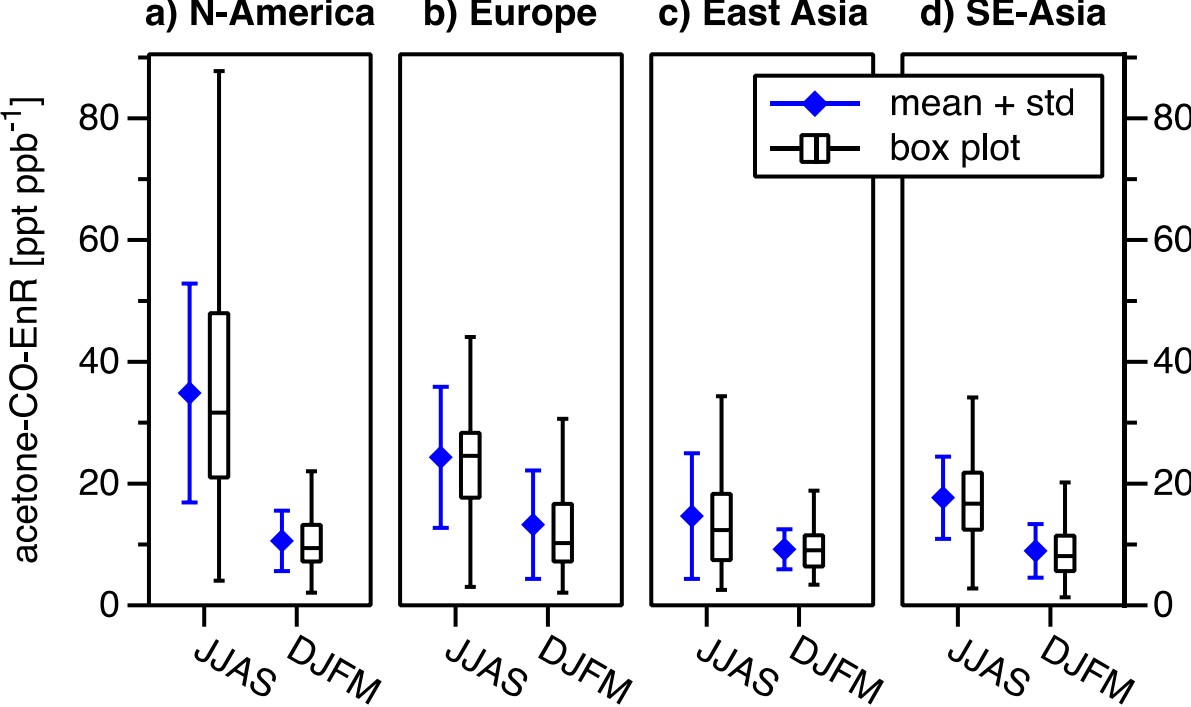

**Figure 8: Mean and standard deviation (std) (blue) and box plots (black) of EnR summer (JJAS) and winter (DJFM) distributions for four selected regions as shown in Fig. 7.**



**Figure 9: North American emission rates of a) acetone and b) CO according to the ECCAD inventory database, averaged over the time period 2005–2010. c) Mean EnRs derived from IAGOS-CARIBIC measurements are compared to ECCAD total emissions volume ratios (TER) of acetone and CO with and without consideration of biomass burning (BB). The dashed lines show sinusoidal functions fitted to the monthly means of EnR and TER (biomass burning excluded).**

[1]**includes propane, α- and β-pinene and methylbutenol as precursors of acetone and ethene, (≥$C_4$)-alkanes, (≥$C_3$)-alkenes and monoterpenes as precursors of CO.**



**Figure 10: Southeast Asia emission rates of a) acetone and b) CO according to the ECCAD database for the time period 2005-2010. c) volume ratio of total emissions of acetone and CO displayed for three different scenarios and compared to EnR derived from IAGOS-CARIBIC in-situ data. A sinusoidal function (dashed line) is fitted to the monthly means of EnR and TER, respectively.**

[1]**includes α- and β-pinene and propane as precursors of acetone and ethene, (≥$C_4$)-alkanes, (≥$C_3$)-alkenes and monoterpenes as precursors of CO.**



| ER | Air mass, location, time | Ref. |
|---|---|---|
| 0.06–0.25 | Biofuel burning | Andreae and Merlet (2001) |
| 0.84 | Vegetation from the southwest U.S.; laboratory experiment | Warneke et al. (2011) |
| 1.2 | Savanna biomass burning | Akagi et al. (2011) |
| 1.6 | Fresh Canadian boreal biomass burning plumes, June–July 2008 | Simpson et al. (2011) |
| 1.7–2.05 | North American wildfires | Friedli et al. (2001) |
| 1.9–4.6 | Savanna and grassland biomass burning | Andreae and Merlet (2001) |
| 2.3–2.7 | Extratropical forest biomass burning | Andreae and Merlet (2001) |
| 1.93 | Vegetation from the southeast U.S.; laboratory experiment | Warneke et al. (2011) |
| 1.94 | Pines spruce; laboratory experiment | Warneke et al. (2011) |
| 2.8 | Boreal forest biomass burning | Akagi et al. (2011) |
| 2.9 | Peatland burning | Akagi et al. (2011) |
| 2.9 | Tropical forest biomass burning | Andreae and Merlet (2001) |
| 2.9 | Charcoal burning | Andreae and Merlet (2001) |
| 2.9 | Residential heating | Kaltsonoudis et al. (2016) |
| 3.0 | Extratropical/Boreal forest biomass burning | Akagi et al. (2011) |
| 3.3 | Tropical forest biomass burning | Akagi et al. (2011) |
| 4.8 | Fresh Savannah fire, Africa | Jost et al. (2003) |
| 5.4 | Savanna grass, laboratory experiment | Holzinger et al. (1999) |
| **2.5 ± 1.3** | **Mean acetone-CO-ER** | |

**Table 1: Literature values of acetone-CO emission ratios (ERs) in ppt ppb$^{-1}$.**



| EnR | Air mass, location, time | Ref. |
|---|---|---|
| 4.7 | Fresh biomass burning plume, summer 2008 | Singh et al. (2010) |
| 5.0 | Biomass burning plumes, Canada, June-July 2008 | Hornbrook et al. (2011) |
| 5.7 | Aged boreal biomass burning plumes from North America, July–August 2011 | Tereszchuk et al. (2013) |
| 6.0 | Biomass burning plumes, California, June-July 2008 | Hornbrook et al. (2011) |
| 6.2 | Aged boreal biomass burning plumes from Siberia, July–August 2011 | Tereszchuk et al. (2013) |
| 6.3 | Aged plumes of Alaskan and Canadian forest fire, July 2004 | De Gouw et al. (2006); |
| 6.6 | Aged Biomass burning plumes, Yucatan, March 2006 | Yokelson et al. (2009) |
| 6.6–22 | Aged biomass burning plumes, free troposphere, Pacific, winter/spring 2001 | Jost et al. (2002) |
| 7.2–10.3 | Biomass burning plumes, South Atlantic, Sept.–October 1992 | Mauzerall et al. (1998) |
| 7.1 | Biomass burning plumes, Canada, June-July 2008 | Hornbrook et al. (2011) |
| 7.5 | Biomass burning plumes, TRACE-P | Singh et al. (2004) |
| 7.7 | Forest Fire Lake Baikal, April 2008 | De Gouw et al. (2009) |
| 9.0 | Asian biomass burning plumes, June-July 2008 | Hornbrook et al. (2011) |
| 10.6 | Aged (1–5 days) biomass burning and urban plumes, summer 2008 | Singh et al. (2010) |
| 11.3 | Fresh Savannah fire plumes (0-125 min plume age), Africa | Jost et al. (2003) |
| 11.7 | Agricultural Fires Kazakhstan, April 2008 | De Gouw et al. (2009) |
| 14.3 | Fresh boreal biomass burning plumes from Siberia, July–August 2011 | Tereszchuk et al. (2013) |
| 16.8 | Fresh boreal biomass burning plumes from North America, July–August 2011 | Tereszchuk et al. (2013) |
| 18 | Aged biomass burning plumes, Crete, August 2001 | Holzinger et al. (2005) |
| 20.4 | Young biomass burning plume, Tanzania, October 2005 (using background VMR over Pacific ocean) | Coheur et al. (2007) |
| **9.9 ± 4.6** | **Mean acetone-CO EnR** | |

**Table 2: Literature values of acetone-CO enhancement ratios (EnRs) in biomass burning plumes in ppt ppb$^{-1}$.**





| EnR | Air mass, location, time | Ref. |
|---|---|---|
| 1.57 | Urban air, Central Eastern China, March-April 2011 | Yuan et al. (2013) |
| 3.18 | Urban air, London, winter 2012 | Valach et al. (2014) |
| 3.4 | Free troposphere, Indian Ocean, February-March 1999 | Reiner et al. (2001) |
| 3.59 | Urban air (with vehicular emissions), Sao Paolo, February–April 2013, | Brito et al. (2015) |
| 5.0 | Mount Tai, China, June 2006 | Inomata et al. (2010) |
| 5.8 | Fresh urban plumes, Eastern U.S., July–August 2004 | Warneke et al. (2007) |
| 6.12 | High-altitude (> 9km), Pacific Ocean, February–March 1994 | McKeen et al. (1997) |
| 13–16 | Marine boundary layer | De Reus et al. (2003) |
| 13.4–17.2 | Ship measurements, Indian Ocean, March 1999 | Whistaler et al. (2002) |
| 14 | Marine boundary layer, Indian Ocean, March 1999 | Reiner et al. (2001); |
| 14.2 | Los Angeles, April–May 2002 | Warneke et al. (2007) |
| 18.3 | Urban plumes, summer 2008 | Singh et al. (2010) |
| 19.5 | Aged high-altitude plumes, Surinam, March 1998 | Andreae et al. (2001) |
| 21–25 | Free troposphere | De Reus et al. (2003) |
| ~22 | Los Angeles Basin, May-June 2010 | Warneke et al. (2012) |
| 30 | Troposphere, Eastern Canada, July–August 1990 | Singh et al. (1994) |
| **12.5 ± 8.6** | **Mean acetone-CO EnR** | |

**Table 3: Literature values of acetone-CO enhancement ratios (EnRs) in air masses unaffected by biomass burning in ppt ppb$^{-1}$.**





| Dilution rate (day$^{-1}$) | Location, time | Ref. |
|---|---|---|
| 0.1 | Atlantic air masses, summer | Arnold et al. (2007) |
| 0.1 | Inside smoke plume, July | Pisso et al. (2009) |
| 0.16 (0.05–0.2) | Biomass burning plume over North Atlantic, summer | Real et al. (2007) |
| 0.24 | Free troposphere, April | Price et al. (2004) |
| 1 | Outside smoke plume, July | Pisso et al. (2009) |
| 1.44 | Planetary boundary layer, South Africa, winter | Igbafe et al. (2006) |
| 1.5 | Mexico city plateau, March | Voss et al. (2010); Shrivastava et al. (2011) |
| 5.5 | Planetary boundary layer, California, summer | Dillon et al. (2002) |
| 4.8–10.3 | Planetary boundary layer, Germany | Kramp and Volz-Thomas (1997) |

**Table 4: Dilution rates in the literature.**





| | | summer | | winter | |
|---|---|---|---|---|---|
| | | **PBL** | **FT** | **PBL** | **FT** |
| **Background [ppb]** | **Acetone** | 2.0 | 0.6 | 0.5 | 0.3 |
| | **CO** | 100 | 70 | 150 | 80 |
| | **Propane** | 0.1 | 0.1 | 1.0 | 0.2 |
| **Enhancement [ppb]** | **Acetone** | 4.2 / 2.0 | | | |
| | **CO** | 140 / 200 | | | |
| | **Propane** | 0.5 | | | |
| **Chemical degradation rate [d⁻¹] (*Lifetime [weeks]*)** | **Acetone** | 0.029 *(5.0)* | 0.049 *(2.9)* | 0.002 *(82.9)* | 0.004 *(37.0)* |
| | **CO** | 0.026 *(5.5)* | 0.039 *(3.6)* | 0.002 *(94.0)* | 0.003 *(49.3)* |
| | **Propane** | 0.111 *(1.3)* | 0.154 *(0.9)* | 0.005 *(26.7)* | 0.009 *(15.2)* |
| **Dilution rate [d⁻¹]** | | 4.80 | 0.10 | 1.44 | 0.05 |

**Table 5: Mixing ratios, reaction and dilution rates used for the simulation of the temporal evolution of acetone-CO EnRs shown in Fig. 3.**



| Season | | Slope/EnR [ppt ppb$^{-1}$] |
|---|---|---|
| **JJAS** | **arithmetic mean $\pm\,\sigma$** | $27.2 \pm 17.0$ |
| | **median** | $22.9$ |
| | **Gaussian line centre $\pm\,\sigma$** | $19.3 \pm 12.9$ |
| | **Number of correlated / all measurements** | $4747 / 12896$ |
| **DJFM** | **arithmetic mean $\pm\,\sigma$** | $11.6 \pm 7.2$ |
| | **median** | $9.4$ |
| | **Gaussian line centre $\pm\,\sigma$** | $8.5 \pm 3.5$ |
| | **Number of correlated / all measurements** | $4137 / 10311$ |

**Table 6: Mean and median values of EnR frequency distributions and centre of the fitted Gaussian distributions.**