# Peer review of "Acetone-CO enhancement ratios in the upper troposphere based on 7 years of CARIBIC data: New insights and estimates of regional acetone fluxes"

_Atmospheric Chemistry and Physics, 2016_

## Referee Comment (RC1) · Anonymous Referee #1 · 22 Sep 2016

Review of doi:10.5194/acp-2016-799

Overview: This paper provides an analysis of acetone-co enhancement ratios observed by IAGOS-CARIBIC flights over the periods 2006-2008 and 2012-2105. A subset of within-plume observations is used along with trajectories to derive emission fluxes, and these observationally based emission fluxes are compared to emission inventories for North America and Southeast Asia. Overall this is a very well written paper with very high quality figures. I recommend the paper for publication in ACP, but encourage the authors to address a few minor things.

[Figure]

Page 8, Line 9: Fischer et al. [2012] and Jacob et al. [2002] include acetone production from higher alkanes (C4 –C5) in addition to propane. The contribution from C4 – C5 alkanes is not negligible, it is on the order of 20% of the contribution from propane.

Page 8, Line 13: It is worth noting that there is a relatively large range of yields reported in the literature for acetone from alpha- and beta- pinene. The yields that were chosen here are on the high end of the ranges. There are published yields for limonene, sabinene, myrecene, delta-3-carene, and ocimene as well. I am not sure if this matters, but perhaps it is worth discussing. See the following references:

Carrasco, N., M. T. Rayez, J. C. Rayez, and J. F. Doussin (2006), Experimental and theoretical study of the reaction of OH radical with sabinene, Phys Chem Chem Phys, 8(27), 3211-3217.

Lee, A., A. H. Goldstein, M. D. Keywood, S. Gao, V. Varutbangkul, R. Bahreini, N. L. Ng, R. C. Flagan, and J. H. Seinfeld (2006), Gas-phase products and secondary aerosol yields from the ozonolysis of ten different terpenes, Journal of Geophysical Research, 111(D7).

Orlando, J. J., B. Nozière, G. S. Tyndall, G. E. Orzechowska, S. E. Paulson, and Y. Rudich (2000), Product studies of the OH- and ozone-initiated oxidation of some monoterpenes, Journal of Geophysical Research, 105(D9), 11561. Reissell, A. (2002), Products of the OH radical- and O3-initiated reactions of myrcene and ocimene, Journal of Geophysical Research, 107(D12).

Reissell, A., C. Harry, S. M. Aschmann, R. Atkinson, and J. Arey (1999), Formation of acetone from the OH radical- and O3-initiated reactions of a series of monoterpenes, Journal of Geophysical Research, 104(D11), 13869.

Vinckier, C., F. Compernolle, A. M. Saleh, N. Van Hoof, and I. Van Hees (1998), Product yields of the alpha-pinene reaction with hydroxyl radicals and the implication on the global emission of trace compounds in the atmosphere., Fresenius environmental

bulletin, 7(5-6), 361-368.

Wisthaler, A., N. R. Jensen, R. Winterhalter, W. Lindinger, and J. Hjorth (2001), Measurements of acetone and other gas phase product yields from the OH-initiated oxidation of terpenes by proton-transfer-reaction mass spectrometry (PTR-MS), Atmospheric Environment, 35, 6181-6191.

Page 13, Lines 5 – 14: Significantly more detail is required here with respect to how the trajectories were used to classify the samples. Did the trajectories simply have to pass through the box for one hour, or the entire trajectory? Was there any altitude requirement. What percentage of the data was classified as linked to one of the four regions? Similarly, the number of measurements (or groups of measurements, i.e. events) associated with each box mean and box plot shown in Figure 8 should be provided with the figure.

Page 14, line 11: How sensitive are the results to choice of a 10 ppb CO enhancement as the cutoff for analysis?

Overall: Do the authors see any differences in the enhancement ratio between the early data 2006 – 2008 and the later data 2012-2015? The emissions of light alkanes over North America are hypothesized to have been growing over this time period, and I am curious as to whether there is a change in the acetone EnRs.

I found only one minor wording fix: Pg 2, line 21: change "remained" to "continues to be"

---

## Referee Comment (RC2) · Anonymous Referee #2 · 4 Nov 2016

The paper of Fischbeck et al. uses a multi annual record of measurements of acetone and CO during IAGOS-CARIBIC in the upper troposphere to estimate emissions of acetone with focus on North America and East Asia. They use observed enhancement ratios of acetone and CO from scatter plots to link them to emission ratios. To focus on distinct air masses instead of spurious incidents of correlated data their method accounts for consecutive data points for slope determination, which thus can be linked to spatially continuous structures. They analyse the tropospheric fraction of data to estimate the emission ratios of Acetone. To deduce this quantity they combine emissions from different inventories for different types of emissions and species. They account

for the chemical evolution and diffusion of the plume during transport by using a box model, which gives the sensitivities of the enhancement ratios to different processes during transport. Based on the model sensitivity studies they define a threshold based on the CO production as indicator for plume ageing and for selecting 'young plumes'. Based on this they estimate seasonally resolved enhancement ratios. They derive fluxes of acetone for North America and south East Asia, which are consistent with the inventories for North America, but derive higher values of acetone fluxes than given in the inventories for East Asia. The paper is in general well written clearly structured and clearly provides a new measurement based approach to estimate fluxes. I therefore recommend it for publication with small changes, which mainly concern the discussion of the potential errors.

What are the contributions to the error for the fluxes (just a statistics) (e.g. for eqn.12)?

What is the range of uncertainty for the curves in Figure 3, when applying e.g. the uncertainties of the main reaction rates? How does this transfer to eqn.12? Further I would appreciate a figure, which shows the derived slopes (or a subset e.g. for Asia) and their variability on the basis of the measured data.

---

## Author Comment (AC1) · 22 Dec 2016

Dear referees,

Thank you very much for your positive feedback and valuable suggestions. We revised the manuscript according to your comments and will discuss the individual points in the following. Your comments are printed in bold, our answers in regular and changes to the manuscript in italic. The revised manuscript is attached to our answers with changes marked in red.

**Referee I**

**Page 8, Line 9: Fischer et al. [2012] and Jacob et al. [2002] include acetone production from higher alkanes (C4 –C5) in addition to propane. The contribution from C4 – C5 alkanes is not negligible, it is on the order of 20% of the contribution from propane.**

In the modelling studies of Jacob et al. (2002) and Fischer et al. (2012), the contribution of propane to the total acetone source is assumed to be in the range of 19 – 33 %, whereas the contribution of C4 – C5 alkanes, namely isobutane and isopentane, is expected to be only 6 – 7 %.

Nevertheless, we decided to include the C4 – C5 alkanes in our calculations, as the effect could also be larger and is worth investigating. In most, if not in all available inventories, isobutane and isopentane are not treated individually. In the MACCity inventory (van der Werf et al., 2006; Lamarque et al., 2010; Granier et al., 2011; Diehl et al., 2012) we use for anthropogenic emissions, isobutane and isopentane are included in the category "butanes and higher alkanes".

Therefore, we estimate the general proportion of those isomers by using the VOC speciation given in Calvert et al. (2009), which is based on the data of Passant (2002). Please note that this data was obtained for the United Kingdom and might not be representative for global emissions. In an effort to find and consider other data, we contacted the authors of another study, but have not get an answer yet. To calculate the secondary acetone production, we use the mean of the respective yields suggested by Jacob et al. (2002) and Pozzer et al. (2010).

As a result, the total acetone fluxes increased on average by ~12 % for North America and by ~8 % for Southeast Asia. In general, the impact is highest in winter (up to 23 % for North America and 10 % for Southeast Asia), when biogenic contributions to the acetone source are small. We conclude that the contribution of C4 – C5 alkanes is indeed not negligible and thank the referee for pointing this out to us.

Changes:

On page 2, line 15, we added: "*The contribution from C4 - C5 alkanes is expected to be 6-7 % (Jacob et al., 2002; Fischer et al., 2012).*"

On page 8, lines 8–9, we changed the sentence to: "*According to Jacob et al. (2002) and Fischer et al. (2012), the three dominant precursors of acetone are propane (13-22 Tg a$^{-1}$ acetone), higher alkanes (4-7 Tg a$^{-1}$ acetone) and monoterpenes (5-6 Tg a$^{-1}$ acetone).*"

On page 8, line 11, we added: "*For isobutane and isopentane, we use the "butanes and higher alkanes" data of MACCity and calculate the proportion of the two species according to the VOC speciation of Passant (2002) and Calvert et al. (2009). The resulting amount of acetone is derived using the means of the yields suggested by Jacob et al. (2002) and Pozzer et al. (2010), which are 0.96 mol mol$^{-1}$ for isobutane and 0.72 mol mol$^{-1}$ for isopentane.*"

On page 18, lines 8-9, we likewise added the consideration of higher alkanes: "*Jacob et al. (2002) and Fischer et al. (2012) assumed that propane is by far the dominant precursor of acetone on a global scale followed by higher alkanes and monoterpenes. Here, we consider propane, isobutane and isopentane, [...]*"

On page 38, Figure 9, and on page 39, Figure 10, the acetone production from isobutane and isopentane was included.

**Page 8, Line 13: It is worth noting that there is a relatively large range of yields reported in the literature for acetone from alpha- and beta- pinene. The yields that were chosen here are on the high end of the ranges. There are published yields for limonene, sabinene, myrcene, delta-3-carene, and ocimene as well. I am not sure if this matters, but perhaps it is worth discussing. See the following references: [...]**

You are right. We chose the yields of Wisthaler et al. (2001), as the authors included secondary acetone formation from the degradation of intermediates in their yields. This might explain why the yields are on the high end of the ranges.

However, considering the literature you refer to, we come to the conclusion that we might have oversimplified our treatment of monoterpenes in general. Consequently, we followed your suggestion and also included the above mentioned monoterpenes. There is no emission inventory in the ECCAD database that differs between the individual monoterpenes, but the partitioning given in Sindelarova et al. (2014) seems to be adequate, as it is of the same authors providing the MEGAN-MACC inventory we use.

Regarding the respective acetone yields from monoterpene oxidation, we refrain from relying on only one study and now consider the available literature when calculating mean yields for each compound and the two main degradation processes (reaction with OH and $O_3$). The two processes are weighted according to the reaction rates. The same was done for methylbutenol (MBO). We should note that we did not include the yields of the work by Vinckier et al. (1998), which you mentioned, as they conducted their experiment at a significantly lower pressure (~2.7 hPa) compared to all other studies, which were performed at atmospheric pressure (Wisthaler et al., 2001). We compiled the yields of all considered studies and our calculations in a separate spreadsheet provided as a supplement.

Although the mean yields for $\alpha$-pinene and $\beta$-pinene (8.9 % and 9.3 %, respectively) are lower than the previously used ones, the additional consideration of the other monoterpene species led to an increase in the calculated amount of acetone produced from the sum of monoterpenes (molar yield of 11.9 % instead of 7.8 %). Assuming global monoterpene emissions

of 95 Tg a$^{-1}$ (Sindelarova et al., 2014), this corresponds to an annual production of 4.3 Tg acetone, which is well in the range given by Jacob et al. (2002) (3–9 Tg) and close to the amount estimated by Fischer et al. (2012) (5 Tg including MBO).

Changes:

5  On page 8, line 11, we included: "*For the monoterpenes, we use the emission data for the sum of monoterpenes from MEGAN-MACC (Sindelarova et al., 2014) and the relative contributions provided in Sindelarova et al. (2014) to calculate the emissions of the following individual monoterpene species: α-pinene, β-pinene, limonene, trans-β-ocimene, myrcene, sabinene and 3-carene. For each species, we derive mean acetone yields based on the available literature. Here, we consider the two main degradation processes of monoterpenes, reaction with OH and O$_3$, and weight the yields*

10  *according to the respective reaction rates (i.e. to the importance of the reaction on the degradation process). All considered yields and calculations are provided as a supplement*."

On page 15, line 9, we revised the acetone bottom-up estimate for North America and assessment thereof with regard to the contributions from isobutane, isopentane and individual monoterpene species: "*This is in very good agreement with the bottom-up estimate of 5.8 Tg a$^{-1}$*" instead of "This is in good agreement with the bottom-up estimate of 5.4 Tg a$^{-1}$".

15  On page 16, lines 21–22, we revised the acetone bottom-up estimate for Southeast Asia accordingly: "*The inventory data for acetone and its precursors suggests a mean annual flux of 149 10$^{-13}$ kg m$^{-2}$ s$^{-1}$ and an annual source of 3.7 Tg a$^{-1}$ for Southeast Asia, which is lower than our estimates, but well within the standard deviation*."

On page 18, lines 9–10, we included the differentiation between the individual monoterpene species: "*Here, we consider propane, isobutane and isopentane, seven individual monoterpene species and methylbutenol (MBO) as precursors of*

20  *acetone*."

On page 18, lines 11–13, we likewise adapted the sentence regarding MBO and adapted the numbers: "*As MBO emission data are not available in the ECCAD-database, we scale the monthly monoterpene surface emissions in a way that annual MBO emissions equal the estimate of 2.2 Tg by Guenther et al. (2012). Based on the available literature, we derive and apply a mean molar acetone yield of 0.46 for the oxidation of MBO. When assuming an instantaneous*

25  *conversion on ground (which is justified for the shorter-lived precursors) MBO oxidation leads to an annual mean acetone production of ~5.6 10$^{-13}$ kg m$^{-2}$ s$^{-1}$, whereas the oxidation of propane (~2.7 10$^{-13}$ kg m$^{-2}$ s$^{-1}$), higher alkanes (~3.4 10$^{-13}$ kg m$^{-2}$ s$^{-1}$) and of monoterpenes (~3.6 10$^{-13}$ kg m$^{-2}$ s$^{-1}$) produce considerably less acetone. Secondary production is largest in summer (~14 10$^{-13}$ kg m$^{-2}$ s$^{-1}$, ~4.1 10$^{-13}$ kg m$^{-2}$ s$^{-1}$, ~3.4 10$^{-13}$ kg m$^{-2}$ s$^{-1}$ and ~9.1 10$^{-13}$ kg m$^{-2}$ s$^{-1}$ respectively)*".

30  On page 19, lines 6–9, we updated the numbers accordingly: "*Biogenic emission dominates throughout the year with a minimum in January (~64 10$^{-13}$ kg m$^{-2}$ s$^{-1}$) and a maximum in May (~154 10$^{-13}$ kg m$^{-2}$ s$^{-1}$). Biomass burning emissions peak in the late dry season (January–April) and contribute to an annual maximum of acetone emissions in March (~208 10$^{-13}$ kg m$^{-2}$ s$^{-1}$) being twice as large as the maximum acetone flux of North America*."

On page 38, Figure 9, the data was updated and the footnote in the caption changed to "*includes propane, isobutane, isopentane, seven monoterpene species and methylbutenol as precursors of acetone and ethene, ($\geq C_4$)-alkanes, ($\geq C_3$)-alkenes and monoterpenes as precursors of CO.*"

On page 39, Figure 10, data and footnote were changed accordingly to: "*includes propane, isobutane, isopentane, seven monoterpene species and methylbutenol as precursors of acetone and ethene, ($\geq C4$)-alkanes, ($\geq C3$)-alkenes and monoterpenes as precursors of CO.*"

**Page 13, Lines 5 – 14: Significantly more detail is required here with respect to how the trajectories were used to classify the samples. Did the trajectories simply have to pass through the box for one hour, or the entire trajectory? Was there any altitude requirement. What percentage of the data was classified as linked to one of the four regions? Similarly, the number of measurements (or groups of measurements, i.e. events) associated with each box mean and box plot shown in Figure 8 should be provided with the figure.**

In a first step, all EnRs, assigned to a trajectory passing the box in any way, have been assigned to the corresponding region. This was done regardless of the trajectory's residence time inside the box. However, when we calculated the (weighted) mean EnR of the box, the individual EnRs were weighted according to the trajectory's residence time inside the box. This means that an EnR assigned to a trajectory, which was within the box for the entire period, has been weighted with a factor of 120 (=5×24 hours). There was no altitude requirement. We have deliberately refrained from such a requirement, because not all transport processes are represented by single trajectory calculations (e.g. convection; cf. Stohl et al., 2002). Therefore, we believe it is not unlikely, that samples assigned to a trajectory, which is at high altitudes for the entire 5 days, had contact with convected air masses from the boundary layer of the underlying region. As the probability of such an incident increases with time, we chose the residence time of the trajectory above the respective region as weighting factor.

We agree with you that the number of underlying events should be given in Figure 8. In this respect, we should also clarify that the EnRs in the underlying distributions have been weighted as described above. This is simply attained by duplicating each EnR for each hour the assigned trajectory was above the region of interest. For the sake of clarity, we decided to provide the numbers of unique (i.e. not duplicated) EnRs and the percentages related to the total number of EnRs in a separate table.

Please note that individual EnRs are assigned to multiple regions, when the trajectory passes through more than one region. For this reason, adding the numbers or percentages horizontally does not result in the global number of EnRs or 100 %. Most EnRs are assigned to North America or Southeast Asia.

| | North America | | Europe | | East Asia | | Southeast Asia | | global | |
|---|---|---|---|---|---|---|---|---|---|---|
| JJAS | 3060 | 15 % | 1218 | 6 % | 1272 | 6 % | 992 | 5 % | 6535 | 32 % |
| DJFM | 1725 | 8 % | 1688 | 8 % | 685 | 3 % | 2255 | 11 % | 7686 | 37 % |
| other months | 2085 | 10 % | 1344 | 7 % | 1343 | 7 % | 2262 | 11 % | 6431 | 31 % |
| all months | 6870 | 33 % | 4250 | 21 % | 3300 | 16 % | 5509 | 27 % | **20652** | **100 %** |

**Table 7: Numbers and percentages of individual EnRs for different months and regions. EnRs are associated with a specific region, when the assigned trajectory passes through the box of the region. For the assignment it does not matter whether only one waypoint or the entire trajectory lies inside the box. Assignment to multiple regions occurs, when the trajectory crosses more than one box. The boxes of the respective regions are indicated in Fig. 8.**

Changes:

On page 13, line 8, we inserted "*In practice, this means that for each hourly waypoint of the trajectory, the assigned EnR is duplicated and associated with the coordinates of the waypoint instead of the sample location.*"

In lines 9–10, "*[EnRs] duplicated this way [...]*" was added.

The three sentences in lines 11–15 have been replaced by a new paragraph describing the source assignment and the underlying distributions of Fig. 8 in more detail: "*The assignment of EnRs to source regions is done as follows: In a first step, longitude-latitude-boxes are defined for the regions of interest. Here, we focus on the four regions North America, Europe, East Asia and Southeast Asia as depicted in Fig. 8. All EnRs with coordinates inside a box, including the duplicates from the trajectory mapping, are assigned to the corresponding region. This means, that an EnR is initially assigned to a region regardless whether the trajectory passed through the box only for 1 hour or the entire 5 days. However, when it comes to averaging over all EnRs of a certain region, we distinguish between such cases. For this purpose, we do not eliminate the duplicates created by the trajectory mapping. Consequently, an EnR associated with a trajectory passing the box only for 1 hour, is represented only once in the regional subset, whereas an EnR with a trajectory staying in the box for 5 days, is represented in the subset 120 (=5×24 hours) times. When averaging over EnRs of a certain region, this naturally leads to a weighting based on the trajectory's residence time above the region. We refrained from an altitude requirement, as not all transport processes are represented by single trajectory calculations (e.g. convection; cf. Stohl et al., 2002). Furthermore, in single trajectory calculations infinitesimally small air parcel are assumed (e.g. Stohl et al., 2002), whereas the volumes of sampled air masses are extended. For these reasons, we believe that even though a trajectory passes a region at high altitudes, there is certain likelihood that the sampled air mass had contact with convected air masses from the boundary layer of the underlying region. As the probability of such an incident increases with time, weighting according to the trajectory's residence time over the region makes most sense to us.*

*In Fig. 9, the weighted mean EnRs and box plots are shown for each source region and the months JJAS and DJFM, respectively. The numbers of the underlying, unique (i.e. non-duplicated) EnRs and percentages of the total number of EnRs are given in Table 7. Please note that single EnRs can be assigned to multiple regions, when the trajectories pass through more than one region. The best coverage is archived for North America (33 %) and Southeast Asia (27 %), which is also why we focus on these regions in the following.*"

The caption of Fig. 7 on page 36 was adapted accordingly: "*Geographical distribution of EnRs duplicated along the hourly waypoints of the assigned 5-day back trajectories.*"

as well as the one of Fig. 8 on page 37: "*In the underlying distributions, individual EnRs have been duplicated along the hourly waypoints of the assigned 5-day back trajectories to consider the residence time of the samples above the region. The numbers of individual, unique EnRs as well as percentages related to the total number of EnRs are given in Table 7.*"

**Page 14, line 11: How sensitive are the results to choice of a 10 ppb CO enhancement as the cutoff for analysis?**

To be strictly accurate, we used 10 times the measurement uncertainty of CO as threshold, but this is on average close to 10 ppb. With regard to this and your question, we repeated the analysis for North America with exactly 10 ppb and 15 ppb as cutoff thresholds. The number of derived EnRs, their monthly means and standard deviations thereof as well as the resulting acetone source estimates are summarized in Table 8, which we add to and address in a dedicated section of the appendix.

Changes:

On page 11, line 12, we added: "*This problem and the sensitivity of results with regard to the chosen threshold are further addressed in section A.2 of the appendix.*"

On page 14, line 10f. we clarified the used cutoff threshold by adding "*10 times the measurement uncertainty of CO*".

[revised manuscript text omitted]

**Overall: Do the authors see any differences in the enhancement ratio between the early data 2006 – 2008 and the later data 2012-2015? The emissions of light alkanes over North America are hypothesized to have been growing over this time period, and I am curious as to whether there is a change in the acetone EnRs.**

The mean of all EnRs in the period 2006–2008 is $(25.0 \pm 19.8)$ ppt ppb$^{-1}$, whereas the one for 2012–2015 is $(26.8 \pm 18.2)$ ppt ppb$^{-1}$. However, these two values are hardly comparable, as samples are not evenly spread throughout the years. Therefore, we also compare the EnRs on a monthly basis:

[Figure]

Again we do not see an evident trend in the data, which could be related to an increase in the emissions of light alkenes, but we also doubt that such an increase would be clearly visible in acetone-CO EnRs derived with our method.

**I found only one minor wording fix: Pg 2, line 21: change "remained" to "continues to be"**

Thank you for pointing out this error. It was corrected in the revised manuscript.

**Referee II**

**What are the contributions to the error for the fluxes (just a statistics) (e.g. for eqn.12)?**

Here, we used the standard deviation of the mean EnR of the respective month and region. Further contributions come from the flux estimates for CO and its precursors, which are based on inventory data and CO yields taken from the literature. Here, we used the same yields as Duncan et al. (2007), who specified Altshuller (1991) as main reference and provided uncertainties for a few, but not for all precursors. Besides the uncertainties in CO yields, which affect the secondarily produced CO, the larger uncertainty probably lies in the inventory fluxes and underlying emission factors thereof. These factors "attempt to relate the quantity of a pollutant released with an activity associated with the release of that pollutant" (Pouliot et al., 2012), but are often based on relatively few samples, which might be not representative on a global level (Duncan et al., 2007) or might be outdated. We are aware that this is an important topic, as the majority of emission estimates depends on these factors and yields, but going into more detail would be out of the scope of this study.

Changes:

Therefore, we added the following statement to section 3.5 on page 8, line 18: "*As uncertainties are not provided for all yields and emission inventory fluxes, we refrain from performing a comprehensive uncertainty analysis. However, considerable uncertainties might exist and estimates based on these data have to be taken with care. In our analysis, at least the statistical uncertainties of fluxes are strongly reduced by averaging over large regions and time periods.*"

**What is the range of uncertainty for the curves in Figure 3, when applying e.g. the uncertainties of the main reaction rates? How does this transfer to eqn.12?**

We absolutely agree with you that the uncertainties should be considered. The uncertainties of the reaction rates $k$ with OH are reported to be approximately 20 % (Atkinson et al., 2004, 2006). For the acetone photolysis rate, we assumed the same (cf. Neumaier et al., 2014). For reasons of clarity, we included the resulting uncertainty ranges only for selected curves in Fig. 3. Derived uncertainties are lowest in wintertime, when reactions are slow, and for the boundary layer, in which dilution dominates over chemical loss. The highest uncertainty is found for large EnR in the summertime free troposphere and an increase or decrease cannot be predicted with reasonable certainty. Nevertheless, the likelihood of an increase in EnR in the absence of secondary acetone production, as discussed in section 4.1, is considerable.

As we do not know the true uncertainties of the measurement-based EnRs and only use means of a large number thereof in Eq. 12, we do not see a direct relation between the discussion and Eq. 12.

On page 9, lines 7–8, we added "*and uncertainties thereof*" and in line 9: "*The latter are reported to be ~20 %. The same was assumed for the acetone photolysis rate (cf. Neumaier et al., 2014).*"

On page 10, line 13, we added "*For reasons of clarity, the range of uncertainty based on the uncertainties of the main reaction rates is provided only for selected curves.*"

On page 11, line 19, we added "*[...], although we have to admit that the range of uncertainty is very large in one case.*"

On page 32, Figure 3, we added "*For selected curves, the range of uncertainty is exemplarily shown in light grey color.*"

On page 44, Table 5, we added the calculated uncertainties of the chemical degradation rates $L$ and lifetimes and exchanged "reaction" by "*chemical loss*" to avoid misunderstandings regarding $L$ and $k$ rates (e.g. $L_{Ac} = k_{Ac}\,[OH] + J_{Ac}$).

**Further I would appreciate a figure, which shows the derived slopes (or a subset e.g. for Asia) and their variability on the basis of the measured data.**

Plotting all measured data (or a regional subset thereof) together with the found correlations would certainly overload the plot. But we do see your point and therefore decided to add a scatter plot (Fig. 3) showing the acetone-CO data and derived slopes of 17 flights, which were selected in a manner to avoid too much data overlap. Although the selection is not necessarily representative of the full dataset, we still believe that the plot gives a good impression of the diversity of EnRs based on temporally and spatially coherent events. In addition, the figure gives a good opportunity to show the difference in results of the standard least squares fit and the Williams-York fit.

Changes:

On page 7, line 25, we added the following explanatory paragraph in addition to the figure: "*In Fig. 3, the differences between the two approaches and fit algorithms are illustrated based on the data of 17 selected flights. The flights were chosen in such a way that larger overlaps of data were avoided. The diversity of the event-based EnRs, ranging from 1.3 to 77.2 ppt ppb$^{-1}$, is clearly visible. Furthermore, it is shown that initial averaging over the total data (classical approach) instead of averaging over the individual EnRs of coherent events makes a difference. The mean of the individual EnRs (18.6 ppt ppb$^{-1}$) is by a factor of 1.9 larger than the slope of the classical approach (9.8 ppt ppb$^{-1}$). In addition, the EnR of the Williamson-York fit (9.8 ppt ppb$^{-1}$) is smaller than the one of the standard least squares fit (11.2 ppt ppb$^{-1}$), as the former puts (more) weight on the small VMRs due to their lower uncertainties. Less weight is put on the large VMRs, in particular the acetone VMRs, as they have larger uncertainties than the CO VMRs.*"

[Figure]

**Figure 3: Scatter plot of tropospheric acetone and CO data of 17 selected CARIBIC flights. Detected correlations are shown as regression lines plotted in the range of the underlying data. Colorcoding denotes to which flight data and correlations belong to. Data, which did not fall into our event-based correlation criteria, are shown as filled black circles regardless of their flight affiliation. The solid black and the grey dashed line represent the results of a Williamson-York fit (EnR: 9.8 ppt ppb⁻¹) and of a standard least-squares fit (EnR: 11.2 ppt ppb⁻¹), respectively, applied to all data of the 17 flights. The minimum, mean and maximum value of the event-based EnRs are 1.3, 18.6 and 77.2 respectively (all values in ppt ppb⁻¹).**

**Changes not related to the referee comments**

While preparing the data of the new Table 7, we discovered a programming error in the code generating the EnR subset for Southeast Asia. One condition had an incorrect sign. The effects of this error are so small that we barely see visible changes in the corrected versions of Figure 8d) and 10c), which have been renumbered to Figure 9d) and 11c) in the revised manuscript. The derived results and messages of the manuscript are not affected at all. We apologize for the incidence and would appreciate it if the necessary correction is approved. Further minor corrections related to this error include:

On page 1, line 26, page 16, line 15 and page 17, line 13: "*(185 ±80) 10⁻¹³ kg m⁻² s⁻¹*" instead of (186 ± 81) 10⁻¹³ kg m⁻² s⁻¹.

On page 16, line 5: "*12.2 ppt ppb⁻¹*" instead of 12.3 ppt ppb⁻¹.

On page 7, line 15, it should read "*temporally and spatially coherent events*" instead of "temporal and spatial [...]".

[revised manuscript text omitted]

---

## Author Comment (AC3) · 22 Dec 2016

Dear referee,

Please find attached a spreadsheet, containing a compilation of the acetone yields from monoterpene and methylbutenol degradation we considered in our reanalysis. The spreadsheet shall be provided as supplement of the revised manuscript.

With best regards

Garlich Fischbeck - on behalf of all authors -

Please also note the supplement to this comment:
http://www.atmos-chem-phys-discuss.net/acp-2016-799/acp-2016-799-AC3-
supplement.zip